# Diagnostic accuracy of antigen detection in urine and molecular assays testing in different clinical samples for the diagnosis of progressive disseminated histoplasmosis in patients living with HIV/AIDS: A prospective multicenter study in Mexico

Areli Martínez-Gamboa[1], María Dolores Niembro-Ortega[1], Pedro Torres-González[1], Janeth Santiago-Cruz[1], Nancy Guadalupe Velázquez-Zavala[1], Andrea Rangel-Cordero[1], Brenda Crabtree-Ramírez[1], Armando Gamboa-Domínguez[2], Edgardo Reyes-Gutiérrez[2], Gustavo Reyes-Terán[3], Víctor Hugo Lozano-Fernandez[3], Víctor Hugo Ahumada-Topete[3], Pedro Martínez-Ayala[4], Marisol Manríquez-Reyes[5], Juan Pablo Ramírez-Hinojosa[6], Patricia Rodríguez-Zulueta[6], Christian Hernández-León[7], Jesús Ruíz-Quiñones[8], Norma Eréndira Rivera-Martínez[9], Alberto Chaparro-Sánchez[10], Jaime Andrade-Villanueva[4], Luz Alicia González-Hernández[4], Sofia Cruz-Martínez[9], Oscar Flores-Barrientos[8], Jesús Enrique Gaytán-Martínez[10], Martín Magaña-Aquino[11], Axel Cervantes-Sánchez[1], Antonio Olivas-Martínez[12], Javier Araujo-Meléndez[11], María del Rocío Reyes-Montes[13], Esperanza Duarte-Escalante[13], María Guadalupe Frías-De León[14], José Antonio Ramírez[13], María Lucia Taylor[13], Alfredo Ponce de León-Garduño[1], José Sifuentes-Osornio[12]*

1 Department of Infectious Diseases, Instituto Nacional de Ciencias Médicas y Nutrición Salvador Zubirán; Tlalpan, Mexico City, Mexico, 2 Department of Pathology, Instituto Nacional de Ciencias Médicas y Nutrición Salvador Zubirán, Tlalpan, Mexico City, Mexico, 3 Centro de Investigación en Enfermedades Infecciosas, Instituto Nacional de Enfermedades Respiratorias Ismael Cosio Villegas; Tlalpan, Mexico City, Mexico, 4 HIV Unit, Hospital Civil de Guadalajara "Fray Antonio Alcalde"; Guadalajara, Jalisco, Mexico, 5 Department of Internal Medicine, Hospital de Alta Especialidad de Veracruz; Veracruz, Veracruz, Mexico, 6 Department of Infectious Diseases, Hospital General Dr. Manuel Gea González; Tlalpan, Mexico City, Mexico, 7 Area of Infectious Diseases, Department of Internal Medicine, Hospital General de Puebla "Dr. Eduardo Vázquez Navarro"; Puebla, Puebla, Mexico, 8 Intensive Care Unit, Department of Internal Medicine, Hospital "Dr. Juan Graham Casasus"; Villahermosa, Tabasco, Mexico, 9 Adult Infectious Diseases Department, Hospital Regional de Alta Especialidad de Oaxaca, HRAEO; San Bartolo Coyotepec, Oaxaca, Mexico, 10 Department of Infectious Diseases, Hospital de Infectología del Centro Médico Nacional "La Raza", Instituto Mexicano del Seguro Social; Azcapotzalco, Mexico City, Mexico, 11 Department of Internal Medicine, Hospital Central Dr. Ignacio Morones Prieto; San Luis Potosí, San Luis Potosí, Mexico, 12 Department of Medicine, Instituto Nacional de Ciencias Médicas y Nutrición Salvador Zubirán; Tlalpan, Mexico City, Mexico, 13 Mycology Unit, Department of Microbiology and Parasitology, Facultad de Medicina, Universidad Nacional Autónoma de México, Mexico City, Mexico, 14 Research unit, Hospital Regional de Alta Especialidad de Ixtapaluca, Mexico State, Mexico

☯ These authors contributed equally to this work.

* sifuentesosornio@gmail.com

## Abstract

### Background

The progressive disseminated histoplasmosis (PDH) has been associated with severe disease and high risk of death among people living with HIV (PLWHIV). Therefore, the purpose

**Data Availability Statement:** All database is available in figshare repository: https://figshare.com/s/26d91c9f2b37dc922a38.

**Funding:** This work was supported by the Consejo Nacional de Ciencia y Tecnología (CONACyT) [FOSISS 2015-1 261482] https://www.conacyt.gob.mx/ to AMG. The funder had no role in study design, data collection and analysis, decision to publish, or preparation of the manuscript.

**Competing interests:** The authors have declared that no competing interests exist.

of this multicenter, prospective, double-blinded study done in ten Mexican hospitals was to determine the diagnostic accuracy of detecting *Histoplasma capsulatum* antigen in urine using the IMMY ALPHA *Histoplasma* EIA kit (IAHE), clarus *Histoplasma* GM Enzyme Immunoassay (cHGEI IMMY) and MiraVista *Histoplasma* Urine Antigen LFA (MVHUALFA); as well as the Hcp100 and 1281-1283$_{220}$SCAR nested PCRs in blood, bone-marrow, tissue biopsies and urine.

## Methodology/Principal findings

We included 415 PLWHIV older than 18 years of age with suspicion of PDH. Using as diagnostic standard recovery of *H. capsulatum* in blood, bone marrow or tissue cultures, or histopathological exam compatible, detected 108 patients (26%, [95%CI, 21.78–30.22]) with proven-PDH. We analyzed 391 urine samples by the IAHE, cHGEI IMMY and MVHUALFA; the sensitivity/specificity values obtained were 67.3% (95% CI, 57.4–76.2) / 96.2% (95% CI, 93.2–98.0) for IAHE, 91.3% (95% CI, 84.2–96.0) / 90.9% (95% CI, 87.0–94.0) for cHGEI IMMY and 90.4% (95% CI, 83.0–95.3) / 92.3% (95% CI, 88.6–95.1) for MVHUALFA.

The Hcp100 nested PCR was performed on 393, 343, 75 and 297, blood, bone marrow, tissue and urine samples respectively; the sensitivity/specificity values obtained were 62.9% (95%CI, 53.3–72.5)/ 89.5% (95%CI, 86.0–93.0), 65.9% (95%CI, 56.0–75.8)/ 89.0% (95%CI, 85.2–92.9), 62.1% (95%CI, 44.4–79.7)/ 82.6% (95%CI, 71.7–93.6) and 34.9% (95%CI, 24.8–46.2)/ 67.3% (95%CI, 60.6–73.5) respectively; and 1281-1283$_{220}$SCAR nested PCR was performed on 392, 344, 75 and 291, respectively; the sensitivity/specificity values obtained were 65.3% (95% CI, 55.9–74.7)/ 58.8% (95%CI, 53.2–64.5), 70.8% (95% CI, 61.3–80.2)/ 52.9% (95%CI, 46.8–59.1), 71.4% (95%CI, 54.7–88.2)/ 40.4% (95%CI, 26.4–54.5) and 18.1% (95%CI, 10.5–28.1)/ 90.4% (95%CI, 85.5–94.0), respectively.

## Conclusions/Significance

The cHGEI IMMY and MVHUALFA tests showed excellent performance for the diagnosis of PDH in PLWHIV. The integration of these tests in clinical laboratories will certainly impact on early diagnosis and treatment.

### Author summary

Histoplasmosis, an infection caused by *Histoplasma capsulatum*, is prevalent in the Americas, it is a common cause of pulmonary acute disease in cave explorers, speleologists, boy scouts and some other people in circumstantial risk, in most of these people the infection is commonly self-limited. However, in people living with HIV (PLWHIV) this infection might be acquired without specific exposition and it behaves like a severe disease with high fever, consumption, septic shock and death. Thus, there is a need for rapid and accurate methods for diagnosis in this population at risk. We tested five different methods for rapid diagnosis (three based on antigen detection in urine and two molecular assays based on PCR amplification, widely used) of disseminated histoplasmosis and we were able to demonstrate that two urine antigen detection tests (clarus *Histoplasma* GM Enzyme Immunoassay kit and MiraVista *Histoplasma* Urine Antigen LFA) showed excellent performance to diagnose of disseminated histoplasmosis in PLWHIV. The antigen detection tests have advantages over the PCR tests, their performance is higher, they are commercial

standardized tests, easy to perform, and provide results in hours, therefore the integration of these tests in clinical laboratories will certainly impact on early diagnosis/treatment and consequently on the outcome of patients.

## Introduction

Histoplasmosis is a fungal infection caused by *Histoplasma capsulatum*. The existence of this microorganism has been reported in many areas around the world, however, the American Continent has been described as endemic to the disease, and Mexico is not the exception [1,2]. In Mexico, HIV infection is a major health problem, until 2019, a total of 210,931 cases of HIV / AIDS had been reported, and 15,653 new cases were diagnosed and notified that same year. [3] In people living with HIV (PLWHIV) residing in endemics areas, histoplasmosis occurs in 2% to 25%, and might be the first manifestation of AIDS in 50% to 75% of them as progressive disseminated histoplasmosis (PDH).[4] The patients at greatest risk are those with <150 CD4 cells/μL.[5]

Overall, the clinical diagnosis of histoplasmosis represents a challenge, since the signs and symptoms are often non-specific, and may be indistinguishable from those of disseminated tuberculosis [6,7] Histoplasmosis and tuberculosis are among the most common AIDS-defining illnesses, both often seen as disseminated infections, causing rapid and fatal evolution in the absence of treatment [8]. Whereas tuberculosis is a well-known disease present in clinical algorithms and in specific public health programs, disseminated histoplasmosis is relatively neglected in the daily practice. In fact, in the Amazonian context among immunosuppressed patients, the incidence and related deaths to histoplasmosis was higher than that of tuberculosis [9].

The gold standard for the diagnosis of histoplasmosis is culture isolation of the fungus or observation of characteristic intracellular yeasts by histopathology. Unfortunately, the sensitivity of the culture is low, in addition, the fungus requires several weeks to grow in standard media culture, and level 3 security laboratories to handle them. Histopathology also shows low sensitivity, is observer dependent, and *Cryptococcus* spp., *Blastomyces dermatitidis*, *Candida glabrata*, *Pneumocystis jirovecii*, *Coccidioides* spp., *Leishmania* spp., *Toxoplasma gondii and Trypanosoma cruzi*, can be confused with *Histoplasma* yeasts.[10] In addition, invasive procedures are required for sampling. Antibody detection tests frequently show false negative results in PLWHIV.[11] The detection of urine antigen is a very promising approach, since results are achieved in a few hours, with high sensitivity, specificity and a remarkable negative predictive value.[12] Some of these tests are developed locally[13], and others are already commercially available for routine use.[14] Two of the commercially available assays were recently validated, one of them (clarus *Histoplasma* GM Enzyme Immunoassay) showed a sensitivity and specificity of 98% and 97%, respectively when tested in urine samples from HIV patients, [15] while the other [MiraVista *Histoplasma* Urine Antigen LFA] showed a sensitivity and specificity of 92% and 94%, respectively when serum samples from patients with HIV and suspected PDH were analyzed. [16]

Among the molecular tests for the diagnosis of histoplasmosis, the Hcp100 (*H. capsulatum* protein 100 kDa) nested PCR has been widely used by different authors and has shown a sensitivity of up to 100% and a specificity of 92·4%.[17] Another recently described PCR directed to the genetic sequence called $1281-1283_{220}$ SCAR in *H. capsulatum*[18], was evaluated together with Hcp100 nested PCR and real-time quantitative PCR (qPCR) for the ITS1 region of the rDNA. The ITS1 marker used in qPCR was the most sensitive in detecting the pathogen,

followed by Hcp100 nested PCR and $1281–1283_{220}$ simplex PCR.[19] However, these markers have not been analyzed in prospective clinical studies or in PLWHIV.

Therefore, the purpose of this study was to determine the diagnostic accuracy of detecting *Histoplasma* antigen in urine using the IMMY ALPHA *Histoplasma* EIA kit (IAHE), clarus *Histoplasma* GM Enzyme Immunoassay (cHGEI IMMY) and MiraVista *Histoplasma* Urine Antigen LFA (MVHUALFA), as well as the Hcp100 and $1281–1283_{220}$ SCAR markers that use nested PCRs for diagnosing PDH among PLWHA compared to the definition of proven-PDH by the European Organization for Research and Treatment of Cancer and the Mycosis Study Group Research and Education Consortium (EORTC/MSGREC)[20]

## Methods

### Ethics statement

The protocol was reviewed and approved by the Human Ethics and Biomedical Research Committees of the coordinator center, Instituto Nacional de Ciencias Médicas y Nutrición Salvador Zubirán (REF: 1626), and by the institutional review board of each center.

The study was conducted according to the principles expressed in the Declaration of Helsinki. Physicians collaborating in the study invited patients with suspected PDH to participate and, after explaining the aims of the study, if agreed, patients (or their relatives in the case of mental impairment or critical illness) signed an informed consent form–as approved by the ethics committee.

We conducted a multicenter, prospective, double-blinded, diagnostic test study from December 2015 to April 2018; ten reference centers distributed in seven states of Mexico participated (eight tertiary care centers and two secondary care general hospitals). Men and women older than 18 years of age, with prior diagnosis of HIV infection and "suspected disseminated histoplasmosis" were consecutively identified, followed up and treated by the study´s infectious diseases physicians from each center as previously described.[21]. For inclusion, at least a blood or a bone marrow culture in media supporting *H. capsulatum* growth and 20 ml of urine were required; according to the on-site physician´s criteria, additional tissue samples from diverse anatomical sites were obtained for culture and histopathology. Patients with incomplete clinical information and those who refused to participate in the study were excluded.

### Clinical definitions

Suspected PDH was defined as any patient with any three of the following: fever, weight loss (>5% of total body weight in the last 6 months), lymphadenopathy, hepatomegaly, splenomegaly, mucous ulcers, skin lesions, gastrointestinal bleeding, diarrhea; bicytopenia or pancytopenia (hemoglobin <10 g/dL, total neutrophils count <1.8 x$10^3$/μL, platelets <100,000/μL), elevation of two times over the normal upper limit of aspartate aminotransferase, lactic dehydrogenase and/or ferritin; radiographic evidence of extrapulmonary involvement (lymphadenopathy, hepatomegaly, splenomegaly, liver and/or spleen abscess, pancreatitis, cholecystitis, epididymitis, gastrointestinal, adrenal, brain, heart, bone, prostate or joint lesions). Proven-PDH in PLWHIV was defined according to the revised definitions of invasive fungal disease from the EORTC/MSGREC.[20] Probable-PDH in PLWHIV was defined as any patient with suspected PDH and a positive *Histoplasma* urine antigen test.

We classified as PDH–negative cases as the absence of *H. capsulatum* growth in cultures or the absence of yeast-like structures in the histopathology exam regardless of whether other opportunistic infections or AIDS defining malignancies were diagnosed.

## Sample management

The samples taken included blood, bone marrow, tissue biopsies and urine. The blood samples were inoculated in BD BACTEC Myco/F Lytic medium for culture and in a vacutainer tube with EDTA to perform DNA extraction for PCRs. The bone marrow samples to perform Wright and Grocott's methenamine silver stains and inoculated in BD BACTEC Myco/F Lytic medium, BD BACTEC Peds Plus/F, Löwenstein-Jensen medium and Sabouraud medium for culture and in a vacutainer tube with EDTA for DNA extraction for PCRs; tissue biopsy samples were collected and transported in Stuart medium to the central laboratory to be inoculated for culture in BD BACTEC Plus Aerobic/F, BD BACTEC Plus Anaerobic/F, Löwenstein-Jensen and Sabouraud, and another part was processed for DNA extraction for PCRs. The tissue biopsy samples could also be collected and transported in formalin for histopathological analysis. The urine samples were collected for PCR amplification and antigen detection by the IAHE (Immuno-Mycologics [IMMY], Norman, OK, United States) according to manufacturer´s instructions. The rest of the urine sample was frozen at -70˚C until August 2019, when it was defrosted to perform the antigen detection test using the cHGEI IMMY (IMMY, Norman) and MVHUALFA (MiraVista Diagnostics IN, USA) according to manufacturer´s instructions. All patients provided a single urine sample for all assays. All cultures were processed as previously described.[21]

Preliminary and definitive results of the cultures and histopathological studies were reported to the patient's physician as soon as they were available to follow up the patients and define the treatment. The results of the molecular and antigen detection tests were not informed, nor were they given any clinical value and were only of the knowledge of the principal investigator of this study. The study was double blind, laboratory technicians responsible for the reading and culture processing were blinded to the result of the rest of the samples, and vice versa. The central laboratory of this study where all the samples were analyzed is accredited by the College of American Pathology for bacterial, fungal and mycobacterial identification and antimicrobial susceptibility testing.

## Antigen detection procedures in urine samples

**Urine *Histoplasma* antigen detection test using the IMMY ALPHA *Histoplasma* EIA kit.**   Urine samples were transported at 4˚C and upon arrival to the central laboratory, frozen at –20˚C and processed for antigen detection in batches every two weeks. The test was performed using the IAHE (IMMY, Norman, OK, United States) following the manufacturer´s instructions. Calibrators and positive and negative controls provided by the manufacturer were included in each trial. These kits were purchased from the manufacturer. The results were categorized as positive and negative, as recommended by the manufacturer. Recently, we reported the performance of this test with urine samples from 288 patients.[21]

**Urine *Histoplasma* antigen detection test using the clarus *Histoplasma* GM Enzyme Immunoassay kit.**   This is an immunoenzymatic sandwich microplate assay which employs monoclonal antibodies to detect *Histoplasma* galactomannan in urine. Urine samples were defrosted and placed at room temperature before being processed according to the manufacturer's instructions, using a standard seven-point antigen curve with concentrations ranging from 0.4 to 25 ng/mL. Positive and negative controls provided by the manufacturer were included in each assay. The optical densities (OD) were read at 450 nm and 620/630 nm, and the *Histoplasma* antigen concentration was calculated based on a seven-point calibration curve using a 4-parameter equation. A positive result was considered when a concentration ≥0.2 ng/mL was obtained, as indicated by the manufacturer.

**Urine *Histoplasma* antigen detection test using MiraVista *Histoplasma* Urine Antigen LFA kit.**   This assay is a lateral flow–based immunoassay, which employs polyclonal antibodies in the direct detection of *Histoplasma* antigen in urine. Urine samples were defrosted and waited until they reached room temperature to process according to the manufacturer's instructions. In short, 100 μL of urine was added to the diluent tube (included in the kit), it was mixed thoroughly and 100 μL of the suspension was taken, which was added to the Lateral Flow device. The reading was done visually, 40 min after the test. The presence of two lines, regardless of its intensity, was considered positive. While a negative result was interpreted when only the control line was observed in the Lateral Flow device. The results were categorized as positive and negative, as recommended by the manufacturer. Controls (positive and negative) were included at the time of the kit was open.

## Sample preparation for molecular assays

**DNA extraction.**   1) In peripheral blood and bone marrow samples. Separation of leukocytes from peripheral blood and bone marrow was performed by density gradient using Ficoll hypaque (HISTOPAQUE -1077, Sigma Aldrich, Burlington, MA, United States). The white cell packet was recovered and washed twice with 1 mL of Dulbecco´s Phosphate Buffered Saline (GIBCO, Thermo Fisher, Scientific Baltics UAB, Vilnius, Lithuania) before proceeding to DNA extraction. For DNA extraction, the cell packet was re-suspended in 180 μL of ATL lysis buffer (Qiagen, Hilden, Germany), 20 μL of proteinase K was added, the mixture was incubated at 56°C for 1 hour. Subsequently, the sample was exposed 5 min to a water bath, followed by three cycles of exposure to liquid nitrogen for 1 min and water bath for 2 min. After the lysis, the sample was left 5 min at room temperature to continue the extraction using the DNeasy Blood & Tissue kit (Qiagen) following the manufacturer's recommendations. 2) In tissue samples. The tissue was macerated in a mortar, the lysate was collected and was suspended in 180 μL ATL lysis buffer (Qiagen) and 20 μl of proteinase K, the sample was incubated at 56°C overnight. DNA extraction was performed following the procedure for exposure to liquid nitrogen and water bath, followed by the DNeasy Blood & Tissue kit (Qiagen) previously mentioned. 3) In urine samples. Samples were defrosted and placed at room temperature (15–25°C) prior to DNA extraction. They were subsequently centrifuged at 4°C at 5000 rpm for 30 min. The supernatant was discarded, and the sediment was used for DNA extraction, which was performed using the ONE–4–ALL Genomic DNA Mini Preps kit (Bio Basic, Amherst, NY, United States) according to the manufacturer's recommendations.

## Molecular assay procedures

**Hcp100 nested PCR assays.**   1) In blood, bone marrow, and tissue samples. The nested PCR assay was conducted as described by Bialek et al.[22] with minor modifications. For external PCR we use the primers HCI and HCII. The reaction was carried out in a final volume of 50 μL with the following reagent concentration: 1X Buffer, 1 mM $MgCl_2$, 100 μM dNTPs (Roche Diagnostics, Germany), and 0·5 pmol of each primer, 1·5 U *Taq* DNA polymerase (Invitrogen, CA, United States) and 10 μL of the DNA. The program used was: 95°C for 5 min, 35 cycles of 94°C for 30 s, 65°C for 30 s, 72°C for 1 min, and a final extension of 72°C for 5 min. With which we obtained a product of 391 bp. Five microliters of the product of the first reaction were taken for nested PCR. The primers HCIII and HCIV that define a 210 bp fragment were used. The reaction was carried out in a final volume of 50 μL and the concentration of the reagents was as follows: 1X Buffer, 1 mM $MgCl_2$, 100 μM dNTPs (Roche Diagnostics), 0·5 pmol of each primer, and 1.5 U *Taq* DNA polymerase (Invitrogen). The amplification program was: 95°C for 5 min, 30 cycles of 94°C for 30 s, 72°C for 1 min and a final extension of

72˚C for 5 min. 2) In urine samples the amplification procedure was conducted primarily according to Bialek, et al.[22] with minor modifications by Taylor et al.[23] The first PCR was carried out in a reaction mixture of 25 μL containing 1X Buffer, 2 mM MgCl$_2$, 200 μM dNTPs, 100 pmol of each primer (HCI and HCII), 1.0 U *Taq* DNA polymerase (Applied Biosystems, Inc., Foster City, CA, United States), and 50–100 ng of DNA. The amplification program was: one cycle at 94˚C for 5 min; 35 cycles at 94˚C for 30 s, 50˚C for 30 s and 72˚C for 1 min, plus a final extension cycle at 72˚C for 5 min. The second reaction was carried out in a volume of 25 μL with the following reagent concentrations: 1X Buffer, 1.8 mM MgCl$_2$, 200 μM dNTPs, 100 pmol. of each primer (HCIII and HCIV), 1U *Taq* DNA polymerase (Applied Biosystems), and 1 μL of the product of the first reaction. The amplification conditions were a cycle at 94˚C for 5 min; 30 cycles at 94˚C for 30 s, 65˚C for 30 s and 72˚C for 1 min, and one final cycle at 72˚C for 5 min. All assays included negative controls (water reagents) and positive controls (DNA extracted from a culture of *H. capsulatum*). The result was considered positive when a band of 391 bp and / or 210 bp was observed in the 1.5% agarose gel, while it was considered negative when these bands were not observed.

**1281−1283$_{220}$ SCAR nested PCR assays in blood, bone marrow, tissue biopsies, and urine.** For the first PCR, the reaction mixture comprised 10 ng of DNA in a 25 μL reaction volume containing 1X Buffer, 200 μM dNTPs (Applied Biosystems), 2 mM MgCl$_2$, 1 U of *Taq* polymerase (Applied Biosystems) and 50 pmol of each oligonucleotide, SCAR$_{220}$F (5´–CAT TGTTGGAGGAACCTGCT–3´) and SCAR$_{220}$R (5´–GAGCTGCAGGATGTTTGTTG–3´). The following program was used for the PCR: one cycle at 94˚C for 3 min, followed by 30 cycles of 95˚C for 1 min, 55˚C for 1 min and 72˚C for 1 min with a final extension step at 72˚C for 5 min. The second PCR reaction was conducted with 2 μL of the product from the first PCR in a total reaction volume of 25 μL containing 1X Buffer, 200 μM dNTPs (Applied Biosystems), 2 mM MgCl$_2$, 1 U of *Taq* polymerase (Applied Biosystems) and 50 pmol of each oligonucleotide, SCAR$_{200}$F (5´–CTCATTACTGTCAACACTGCGG–3´) and SCAR$_{200}$R (5´–GC TGCAGGATGTTTGTTGATGT–3´) under the same cycling conditions of the first round of amplification, except that 60˚C for 1 min was used for annealing the template.

**Amplification of β-globin gene (internal control).** To validate the presence of amplifiable DNA and the absence of inhibitory substances, a PCR was performed using GH20 (5´–G AAGAGCCAAGGACAGGTAC–3´) and PCO4 (5´–CAACTTCATCCACGTTCACC–3´) primers.[24] The reaction was carried out in a final volume of 50 μL with the following reagent concentration: 1X Buffer, 1 mM MgCl$_2$, 100 μM dNTP´s, 0·5 pmol of each primer, 1·5 U *Taq* DNA polymerase (Invitrogen) and 2 μL of the DNA. The program used was: 95˚C for 7 min, 35 cycles of 94˚C for 1 min, 50˚C for 30 s, 72˚C for 45 s, and a final extension of 72˚C for 7 min. With which we obtained a product of 268 bp. A negative control (reagents plus water) and a positive control (reagents plus human DNA) were included in each experiment.

## Statistical analysis

We described and compared the patient's characteristics and clinical variables between patients with proven-PDH and patients PDH negative. We used median and interquartile range (IQR) for continuous variables and frequencies and percentages for categorical variables and Mann-Whitney U and Pearson's X2 or Fisher´s exact test as statistical tests accordingly. A *p*-value ≤ 0.05 was determined as significant. We calculated the sensitivity, specificity, positive predictive value (PPV), negative predictive value (NPV), positive and negative likelihood ratios, with their respective 95% confidence intervals (95% CI) of the antigen urine tests and molecular tests, using as gold standard the cases classified as proven-PDH by culture and/or histopathology. The Kappa index and its respective 95% CI were

used to determine the agreement between urine *Histoplasma* antigen detection tests. We use STATA 11.0 software (StataCorp LLC, College Station, TX, United States) to perform the statistical analysis.

## Results

During the study period, 444 patients potentially eligible to participate were seen; 25 were excluded for not complying with the definition of suspected PDH, two with incomplete clinical information, one refused to sign the informed consent and one with no clinical samples available. Then, we included 415 patients for the analysis; 108/415 (26%) patients with proven-PDH, in 30/108 (28%) the diagnosis was done by identification of *Histoplasma* by both culture and histopathology, in 65/108 (60%) by isolation of *Histoplasma* in at least one culture, and in 13/108 (12%) by histopathology; and 41 with probable PDH (Fig 1). The relevant clinical characteristics of the patients included in the study are described in Table 1. Among the 266 patients without PDH, we found 48 (18%) mycobacterial infections (22 by *Mycobacterium tuberculosis*, 25 by *M. avium* complex, and one case by *M. tuberculosis* and *M. avium* complex infection); 29 patients (10.9%) with other diagnosis (17 with bacteremia / fungemia [blood

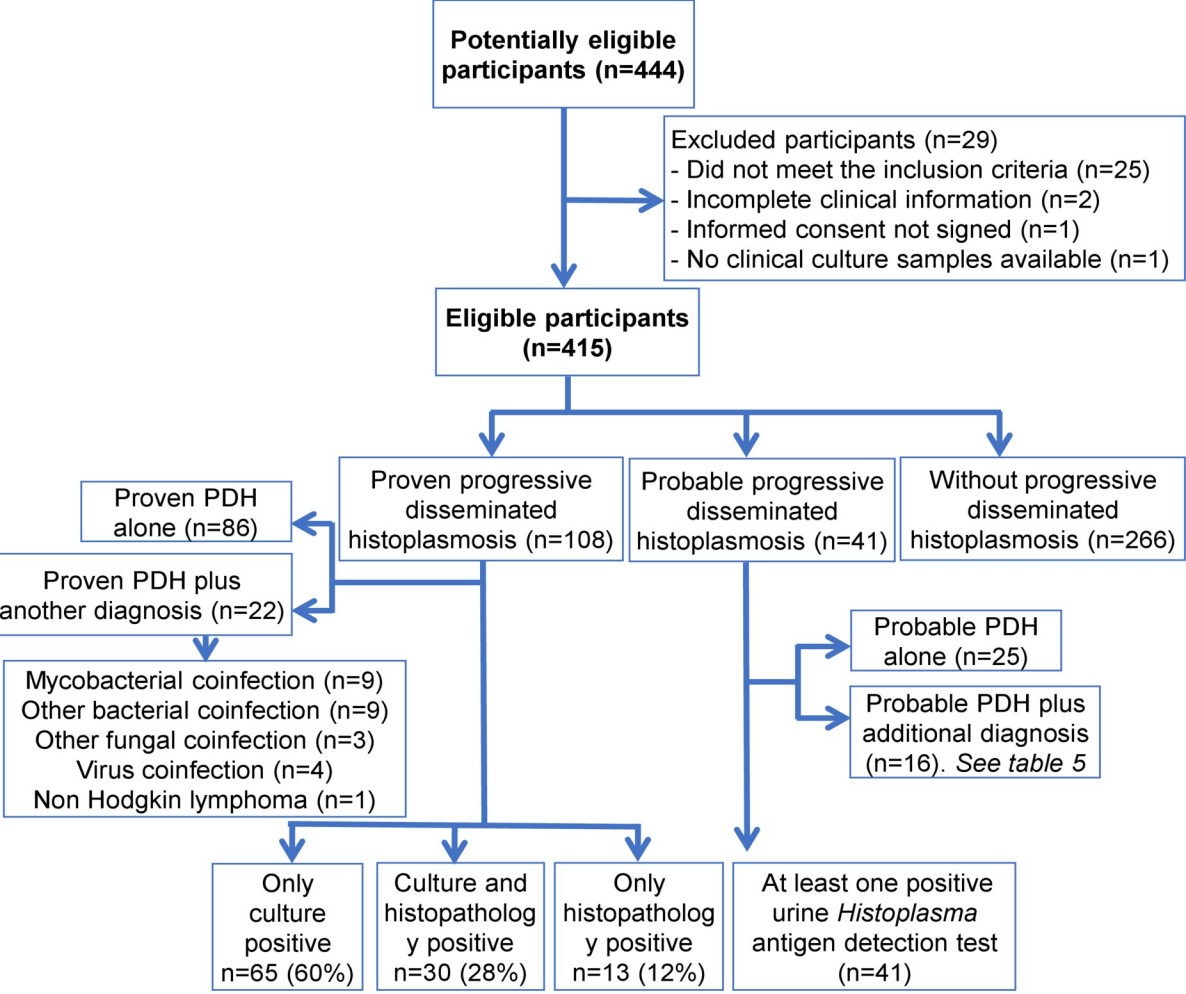

**Fig 1. Study Flowchart.** Abbreviation: PDH = Progressive disseminated histoplasmosis.

**Table 1. Clinical characteristics of the patients included in the study.**

| Demographic characteristic | Proven disseminated histoplasmosis (n = 108) | Probable histoplasmosis (n = 41) | Without PDH (n = 266) |
|---|---|---|---|
| Age, median (IQR) | 33 (11) | 36 (16) | 36 (14) |
| Male, n (%) | 90 (83.3) | 37 (90.2) | 233 (87.6) |
| Risk occupation for histoplasmosis, n (%) | 15 (13.9) | 5 (12.2) | 34 (12.8) |
| Warehouse exposure, n (%) | 14 (13) | 7 (17.1) | 72 (27.1) |
| **HIV Diagnosis** | | | |
| Time with known HIV diagnosis, (months) median (IQR) | 3 (44) | 2 (14) | 3 (43) |
| CD4 count (cells/ μL) median (IQR) | 25.5 (69) | 40.5 (71) | 36.5 (72) |
| Log HIV Viral load, median (IQR) | 5.18 (5.60) | 5.57 (6.15) | 5.34 (5.72) |
| CD4+ count < 100 cells/μL, n (%) | 70 (64.8) | 31 (75.6) | 203 (76.3) |
| ART naïve, n (%) | 62 (57.4) | 23 (56.1) | 142 (53.4) |
| **Signs and symptoms** | | | |
| Fever, n (%) | 107 (99.1) | 41 (100) | 246 (92.5) |
| Weight loss, n (%) | 98 (90.7) | 39 (95.1) | 235 (88.3) |
| Hepatomegaly, n (%) | 71 (65.7) | 27 (65.9) | 193 (72.6) |
| Cough, n (%) | 65 (60.2) | 31 (75.6) | 158 (59.4) |
| Dyspnea, n (%) | 59 (54.6) | 25 (61) | 135 (50.8) |
| Diarrhea, n (%) | 55 (50.9) | 24 (58.5) | 125 (47) |
| Lymphadenopathy, n (%) | 54 (50) | 30 (73.2) | 177 (66.5) |
| Splenomegaly, n (%) | 54 (50) | 15 (36.6) | 155 (58.3) |
| Skin lesions, n (%) | 35 (32.4) | 8 (19.5) | 55 (20.7) |
| Vomit, n (%) | 30 (27.8) | 9 (22) | 48 (18) |
| Mucous ulcers, n (%) | 12 (11.1) | 4 (9.8) | 36 (13.5) |
| Gastrointestinal bleeding, n (%) | 11 (10.2) | 4 (9.8) | 35 (13.2) |
| **Laboratory abnormalities** | | | |
| Anemia, n (%) | 66 (61.1) | 27 (65.9) | 145 (54.5) |
| Thrombocytopenia, n (%) | 58 (53.7) | 20 (48.8) | 82 (30.8) |
| Neutropenia, n (%) | 35 (32.4) | 13 (31.7) | 79 (29.7) |
| Platelets count (cells x1000/mm$^3$), median (IQR) | 90 (135) | 108 (187) | 158 (179) |
| AST elevation 2x NUL, n (%) | 63 (58.3) | 15 (36.6) | 67 (25.2) |
| AST (IU/L), median (IQR) | 107.5 (186) | 55.5 (83) | 46 (57) |
| Alkaline phosphatase (mg/dL), median (IQR) | 223 (309) | 164 (149) | 124 (155) |
| LDH elevation 2x NUL, n (%) | 71 (65.7) | 12 (29.3) | 44 (16.5) |
| LDH > 1000 mg/dL, n (%) | 46 (42.6) | 4 (9.8) | 16 (6) |
| LDH (mg/dL), median (IQR) | 969.5 (1682) | 365 (514) | 302 (301) |
| Sodium (mEq/L), median (IQR) | 131 (6) | 133 (9) | 132 (8) |
| **Radiographic findings** | | | |
| Thorax image | | | |
| Mediastinal adenopathies, n (%) | 13 (12) | 12 (29.3) | 67 (25.2) |
| Tree in bud infiltrate, n (%) | 11 (10.2) | 5 (12.2) | 52 (19.5) |
| Extrapulmonary image | | | |
| Hepatomegaly, n (%) | 40 (37) | 23 (56.1) | 161 (60.5) |
| Splenomegaly, n (%) | 34 (31.5) | 14 (34.1) | 138 (51.9) |
| Neck adenopathies, n (%) | 12 (11.1) | 11 (26.8) | 50 (18.8) |
| Retroperitoneal adenopathies, n (%) | 9 (8.3) | 9 (22) | 39 (14.7) |
| Abdominal adenopathies, n (%) | 8 (7.4) | 5 (12.2) | 31 (11.7) |
| Inguinal adenopathies, n (%) | 2 (1.9) | 6 (14.6) | 14 (5.3) |

*(Continued)*

**Table 1.** (Continued)

| Demographic characteristic | Proven disseminated histoplasmosis (n = 108) | Probable histoplasmosis (n = 41) | Without PDH (n = 266) |
|---|---|---|---|
| Pericardial effusion, n (%) | - | - | 2 (0.8) |
| Mortality | 34/76 (44.7) | 14/34 (41.2) | 64/240 (26.7) |

Abbreviation: PDH = Progressive disseminated histoplasmosis. ART = Antiretroviral treatment. AST = Aspartate aminotransferase. NUL = Normal upper limit. LDH = Lactic dehydrogenase.

and/or bone marrow positive cultures], and 12 with histopathology diagnosis of cytomegalovirus, Kaposi sarcoma, lymphoma and/or HIV lymphadenopathy) and in 189 (71.1%) the diagnosis was not established.

## Diagnostic accuracy of the antigen detection procedures in urine samples

Of the 415 patients, we analyzed 391 urine samples (one per patient) by the IAHE, cHGEI IMMY and MVHUALFA assays. Twenty-six percent (104/391) of the samples analyzed by the three antigen detection tests were proven-PDH cases. The IAHE test detected 70 (67.3%) of them as positive; while 11/ 287 cases without proven-PDH, according to the gold standard, resulted as positive. The cHGEI IMMY test detected 95/104 (91.3%) as positive; while 26/287 cases without proven-PDH were detected as positive. The range of antigen concentration in the cases detected as positive by cHGEI IMMY was 0.43998 ng/mL–>25 ng/mL, and the median was >25 ng/mL. The MVHUALFA test detected 94/104 (90.4%) as positive; while 22/ 287 cases without proven-PDH were detected as positive. The sensitivity/specificity values were 67.3% (95% CI, 57.4–76.2)/ 96.2% (95% CI, 93.2–98.0) for IAHE, 91.3% (95% CI, 84.2–96.0)/ 90.9% (95% CI, 87.0–94.0) for cHGEI IMMY and 90.4% (95% CI, 83.0–95.3)/ 92.3% (95% CI, 88.6–95.1) for MVHUALFA. The sensitivity, specificity, PPV, NPV, LR + and LR-data from all tests are shown in Table 2. More descriptive results are shown in Fig 2. The Fig 3 shows the distribution of the 391 samples analyzed with the three urine *Histoplasma* antigen detection tests. The agreement between the cHGEI IMMY and MVHUALFA was of 0.85 (95% CI, 0.84–0.87).

## Discrepant results between antigen detection tests and gold standard

**False negatives results.** Among patients with proven-PDH, we found seven that had a negative result in the three urine *Histoplasma* antigen detection tests, two were diagnosed by culture only, and the other five by histopathology see Table 3; three that had a negative result in two tests, (one with cHGEI IMMY/MVHUALFA negative, one with IAHE/MVHUALFA negative, and one with IAHE/cHGEI IMMY negative); and 26 had a negative result in one test (25 with IAHE, and one with MVHUALFA).

## Cases with probable progressive disseminated histoplasmosis

Among patients without proven-PDH, 41 presented at least one positive antigen test (probable cases). In 25 of them no other diagnosis was defined (Table 4), while in 16 we found additional diagnoses (Table 5).

From the 41 patients, 26 had only one positive urine *Histoplasma* antigen detection test (six with IAHE positive, 11 with cHGEI IMMY positive, nine MVHUALFA positive). Twelve patients presented two urine antigen positive tests (two patients with IAHE/cHGEI IMMY,

**Table 2. Performance evaluation of three urine *Histoplasma* antigen detection tests and two molecular tests for the diagnosis of progressive disseminated histoplasmosis in people living with human immunodeficiency virus.**

| | Urine antigen | | | Hcp100 nested PCR | | | | SCAR $1281-1283_{220}$ nested PCR | | | |
|---|---|---|---|---|---|---|---|---|---|---|---|
| | IAHE (n = 391) | cHGEI IMMY (n = 391 | MVHUALFA (n = 391) | Blood (n = 393) | Bone marrow (n = 343) | Tissue (n = 75) | Urine (n = 297) | Blood (n = 392) | Bone marrow (n = 344) | Tissue (n = 75) | Urine (n = 291) |
| Sensitivity % (95% CI) | 67.3 (57.4–76.2) | 91.3 (84.2–96.0) | 90.4 (83.0–95.3) | 62.9 (53.3–72.5) | 65.9 (56.0–75.8) | 62.1 (44.4–79.7) | 34.9 (24.8–46.2) | 65.3 (55.9–74.7) | 70.8 (61.3–80.2) | 71.4 (54.7–88.2) | 18.1 (10.5–28.1) |
| Specificity % (95% CI) | 96.2 (93.2–98.0) | 90.9 (87.0–94.0) | 92.3 (88.6–95.1) | 89.5 (86.0–93.0) | 89.0 (85.2–92.9) | 82.6 (71.7–93.6) | 67.3 (60.6–73.5) | 58.8 (53.2–64.5) | 52.9 (46.8–59.1) | 40.4 (26.4–54.5) | 90.4 (85.5–94.0) |
| Accuracy % (95% CI) | 88.5 (84.9–91.5) | 91.0 (87.7–93.6) | 91.8 (88.6–94.3) | 82.9 (78.8–86.5) | 83.0 (78.7–86.9) | 74.6 (63.3–84.0) | 58.2 (52.4–63.9) | 60.4 (55.4–65.3) | 57.5 (52.1–62.8) | 52.0 (40.1–63.7) | 69.7 (64.1–74.9) |
| PPV % (95% CI) | 86.4 (77.0–93.0) | 78.5 (70.1–85.4) | 81.0 (72.7–87.7) | 66.3 (56.6–76.0) | 67.4 (57.5–77.3) | 69.2 (51.5–87.0) | 29.3 (20.6–39.3) | 34.6 (27.7–41.4) | 34.4 (27.5–41.3) | 41.7 (27.7–55.6) | 42.9 (26.3–60.7) |
| NPV % (95% CI) | 89.0 (85.0–92.3) | 96.6 (93.7–98.4) | 96.3 (93.4–98.2) | 88.0 (84.4–91.7) | 88.3 (84.4–92.3) | 77.6 (65.9–89.2) | 72.7 (66.0–78.8) | 83.6 (78.5–88.6) | 83.9 (78.2–89.5) | 70.4 (53.1–87.6) | 73.4 (67.6–78.7) |
| LR+ (95% CI) | 17.7 (9.8–33.0) | 10.0 (7.5–12.6) | 11.8 (8.5–15.4) | 6.00 (4.16–8.66) | 6.00 (4.10–8.78) | 3.57 (1.79–7.12) | 1.1 (0.7–1.5) | 1.59 (1.30–1.94) | 1.50 (1.25–1.81) | 1.20 (0.86–1.67) | 1.9 (1.0–3.6) |
| LR- (95% CI) | 0.3 (0.3–0.4) | 0.1 (0.0–0.2) | 0.1 (0.1–0.2) | 0.41 (0.32–0.54) | 0.38 (0.29–0.51) | 0.46 (0.28–0.75) | 1.0 (0.8–1.2) | 0.59 (0.44–0.79) | 0.55 (0.39–0.78) | 0.71 (0.36–1.40) | 0.9 (0.8–1.0) |

Abbreviation: CI = Confidence Interval. IAHE = IMMY ALPHA *Histoplasma* EIA kit. cHGEI IMMY = clarus *Histoplasma* GM Enzyme Immunoassay.

MVHUALFA = MiraVista *Histoplasma* Urine Antigen LFA.

and 10 with cHGEI IMMY/MVHUALFA). Finally, three patients had positive results in the three tests.

## Diagnostic accuracy of the molecular assays in blood, bone marrow, tissue biopsies and urine samples

Of the 415 patients, the Hcp100 nested PCR was performed on 393, 343, 75 and 297, blood, bone marrow, tissue and urine samples (one per patient) respectively. Twenty-five percent (97/393) of the blood samples analyzed were proven-PDH cases, 61 (62.9%) of them had a positive test. Thirty-one of the 296 cases without PDH resulted false positive. Twenty-six percent (88/343) of the bone marrow samples came from proven-PDH cases. Fifty-eight (65.9%) of them had a positive test. Of the 255 cases without PDH, this test detected 28 as false positive. Thirty-nine percent (29/75) of the tissue samples were cases of proven-PDH. Eighteen (62.1%) of them had a positive test. Of the 46 cases without PDH, eight were detected as false positive. Twenty-eight percent (83/297) of the urine samples were obtained from proven-PDH cases. Twenty-nine (34.9%) of them had a positive test. Of the 214 cases without PDH, the test detected 70 as false positive, see Fig 4.

Of the 415 patients, we analyzed 392, 344, 75 and 291, blood, bone marrow, tissue and urine samples (one per patient) respectively by $1281-1283_{220}$ SCAR nested PCR. Twenty-five percent (98/392) of the blood samples were cases of proven-PDH. Sixty-four (65.3%) of them had a positive test. Of the 294 cases without PDH, 121 were detected as false positive. Twenty-six percent (89/344) of the bone marrow samples were cases of proven-PDH. Sixty-three

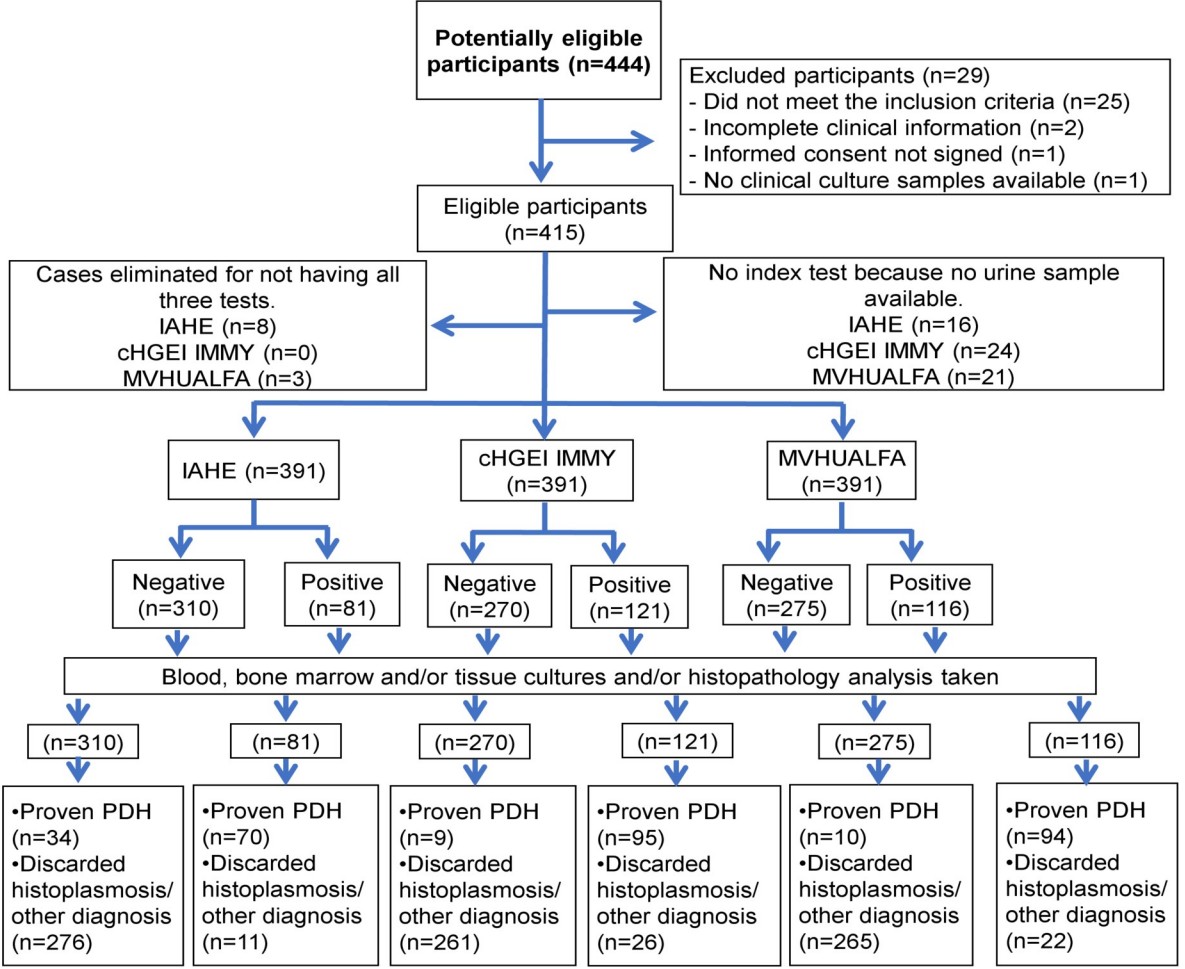

**Fig 2. Flowchart of urine samples submitted to the three urine *Histoplasma* antigen detection tests.** Abbreviation: IAHE = IMMY ALPHA *Histoplasma* EIA kit. cHGEI IMMY = clarus *Histoplasma* GM Enzyme Immunoassay. MVHUALFA = MiraVista *Histoplasma* Urine Antigen LFA. PDH = Progressive disseminated histoplasmosis. No inconclusive results were found.

(70.8%) of them had a positive test. Of the 255 cases without PDH, the test detected 120 as false positive. Thirty-seven percent (28/75) of the tissue samples were cases of proven-PDH. Twenty (71.4%) of them had a positive test. Of the 47 cases without PDH, 28 were detected as false positive for the test. Twenty nine percent (83/291) of the urine samples belonged to patients of proven-PDH. Fifteen (18.1%) of them had a positive test. Of the 208 cases without PDH, the test detected 20 as false positive, see Fig 5. Sensitivity, specificity, PPV, NPV, LR + and LR- data for all test are shown in Table 3.

## Discussion

The data presented in this report show that the new methods (cHGEI IMMY and MVHUALFA) for the detection of *Histoplasma* antigen in urine are excellent tools for diagnosing PDH in PLWHIV. These methods are based on simple and achievable procedures for most of the clinical laboratories and both are far superior to molecular methods (Hcp100 and -1281-1283220 SCAR markers that use nested PCRs), due to the high values of sensitivity/specificity, 91.3% (95% CI, 84.2–96.0) / 90.9% (95% CI, 87.0–94.0) for cHGEI IMMY and 90.4%

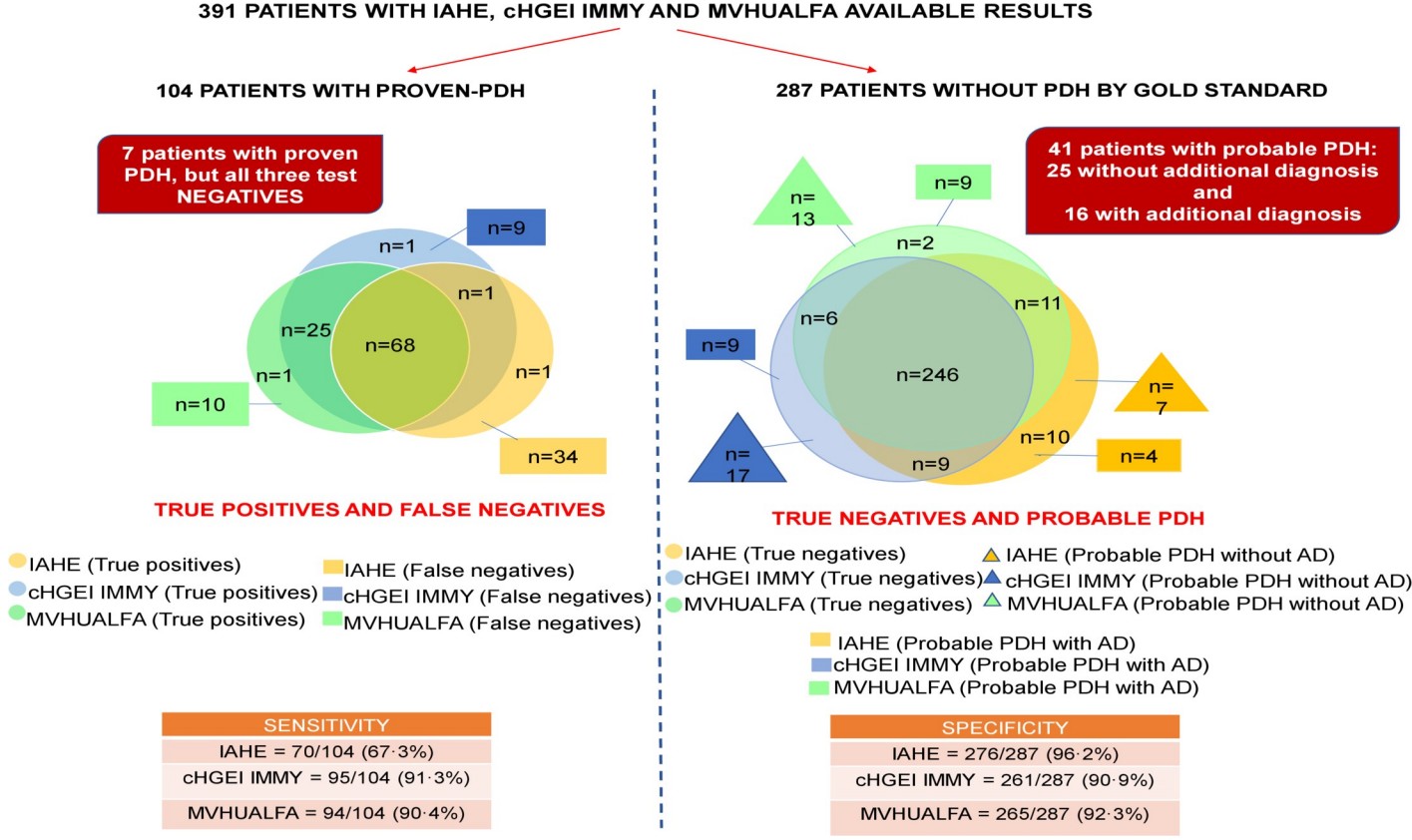

**Fig 3. Distribution of the 391 samples analyzed with the three urine *Histoplasma* antigen detection tests.** Abbreviation: IAHE = IMMY ALPHA *Histoplasma* EIA kit. cHGEI IMMY = clarus *Histoplasma* GM Enzyme Immunoassay. MVHUALFA = MiraVista *Histoplasma* Urine Antigen LFA. AD = Additional diagnosis.

(95% CI, 83.0–95.3) / 92.3% (95% CI, 88.6–95.1) for MVHUALFA, obtained in this clinical trial.

Additional advantages of the MVHUALFA are the possibility of doing a single test without waiting until a certain quota is filled with a plate or platform, the ease of performing the test, practically next to the patient's bed, which allows having the result in minutes and at a very reasonable cost and, finally, the possibility of execution by personnel with minimal training. In this cohort study with the highest number of PDH cases among PLWHIV, we detected 149 patients with histoplasmosis (35.9% of prevalence), 108 of them with proven-PDH and 41 with probable-PDH (based on any positive urine *Histoplasma* antigen test). This prevalence of histoplasmosis among PLWHIV seems to be higher than that reported from Guatemala (8%) and Colombia (23%),[15] which gives an additional support to these findings.

In the late 1980s MiraVista introduced the detection of *Histoplasma* antigen in urine and serum samples as a diagnostic tool for histoplasmosis.[13] This test has shown sensitivity values of 95% to 100% in urine and 92% to 100% in serum.[11,25] However, it is performed at MiraVista laboratories in Indianapolis USA, which delays the availability of the results, with the consequent delay of the appropriate treatment and great risk of complications and death. On the other hand, IMMY launched the first screening test for *Histoplasma* antigen in urine (IAHE), through an ELISA test using polyclonal antibodies approved by FDA in 2007.[14] In a recent publication, we reported a sensitivity and specificity of 67·1% and 97.5% respectively, for the diagnosis of PDH in PLWHIV.[21] Recently, this test was discontinued, since the manufacturer delivered the new platform, cHGEI IMMY.

**Table 3. Clinical and laboratory characteristics of proven-PDH cases with a negative result by all three urine *Histoplasma* antigen detection tests.**

| | Age/ Gender | Time since HIV diagnosis /Relevant medical history | CD4 + (cells/ mL)/ Plasma RNA HIV (copies/mL)/ Antiretroviral treatment | Signs and symptoms | Radiographic Pulmonary and extrapulmo-nary involvement | Empirical antifun-gal before samples | Sites sampled for Culture / Pathology | Final diagnosis/ Treatment/ Evolution | Hcp100 nested PCR | 1281 −1283$_{220}$ SCAR nested PCR |
|---|---|---|---|---|---|---|---|---|---|---|
| 1 | 31/ Male | 2 mo/History of esophageal candida | 25/ 26,133/ Yes (TDF + FTC + EFV) | Weight loss, nausea, vomit, diarrhea, fever, peripheral adenopathies, hepatomegaly, splenomegaly, acute renal failure, anemia, thrombocytopenia LDH>x2 NUL, ferritin elevation | Yes (tree in bud pattern and cavitary lesions; mediastinal adenopathies, hepato-splenomegaly) | FLZ (<72 h) | Blood culture: negative BM culture: *H. capsulatum* Bone biopsy pathology: negative | Proven PDH/ Unknown/ Death due to PDH | ▪Blood: + ▪BM: + ▪Tissue: NA ▪Urine: - | ▪Blood: + ▪BM: + ▪Tissue: NA ▪Urine: - |
| 2 | 33 / Male | 49 mo/History of Kaposi´s sarcoma and syphilis; chemotherapy | 111/ 88,000/ Yes (TDF + ABC + 3TC + RAL) | Weight loss, nausea, vomit, diarrhea, fever, anemia, thrombocytopenia, AST >x2 NUL | No | AMB (<72 h) | Blood culture: *H. capsulatum* BM culture: *H. capsulatum* and *Salmonella* sp. Group D Biopsy pathology: NA | Proven PDH and *Salmonella sp.* Group D infection/ Unknown/ Death due to PDH | ▪Blood: + ▪BM: - ▪Tissue: NA ▪Urine: - | ▪Blood: + ▪BM: + ▪Tissue: NA ▪Urine: - |
| 3 | 19 / Female | 3 mo/History of *M. tuberculosis* infection, PCP, corticoste-roids consumption | 59/ 2,170,000/ Yes (TDF + FTC + EFV) | Weight loss, dyspnea, cough, fever, anemia, LDH >x2 NUL | No | No | Blood culture: negative BM culture: negative BM biopsy pathology: *H. capsulatum* | Proven PDH/ Unknown/ Death due to PDH | ▪Blood: - ▪BM: - ▪Tissue: NA ▪Urine: NA | ▪Blood: + ▪BM: + ▪Tissue: NA ▪Urine: NA |
| 4 | 34 / Male | 68 mo/History of Kaposi´s sarcoma, corticoste-roids consumption | Unknown/ Unknown/ Yes (TDF + DRV + RTV + RAL) | Weight loss, cough, fever, peripheral adenopathies, hepatomegaly, splenomegaly, cutaneous lesions, LDH >x2 NUL | Yes (Neck, mediastinal, abdominal adenopathies; hepatomegaly splenomegaly) | ITZ (>72 h) | Blood culture: MAC BM culture: MAC BM biopsy pathology: *H. capsulatum* | Proven PDH and MAC infection/ Unknown/ Death due to PDH | Blood: - BM: - Tissue: NA Urine: - | Blood: + BM: + Tissue:NA Urine: - |
| 5 | 29 / Male | Simultaneous/ None | 3/ 233, 163/ None | Weight loss, fever, peripheral adenopathies, mucosal ulcers, cutaneous lesions, anemia | Yes (micronodular infiltrate; neck adenopathies) | No | Blood culture: negative BM culture: negative Bone biopsy pathology: *H. capsulatum* | Proven PDH/ Unknown/ Death due to PDH | ▪Blood: - ▪BM: - ▪Tissue: NA ▪Urine: - | ▪Blood: - ▪BM: - ▪Tissue: NA ▪Urine: - |

(*Continued*)

**Table 3.** (Continued)

| | Age/ Gender | Time since HIV diagnosis /Relevant medical history | CD4 + (cells/ mL)/ Plasma RNA HIV (copies/mL)/ Antiretroviral treatment | Signs and symptoms | Radiographic Pulmonary and extrapulmo-nary involvement | Empirical antifun-gal before samples | Sites sampled for Culture / Pathology | Final diagnosis/ Treatment/ Evolution | Hcp100 nested PCR | 1281 −1283$_{220}$ SCAR nested PCR |
|---|---|---|---|---|---|---|---|---|---|---|
| 6 | 24 / Male | 27 mo/none | 120/ 40,889/ None | Weight loss, fever, mucosal ulcers, cutaneous lesions, anemia | No | No | Blood culture: negative BM culture: negative Skin biopsy culture: negative Skin biopsy pathology: *H. capsulatum* | Proven PDH/ Unknown/ Unknown | ∎Blood: - ∎BM: - ∎Tissue: - ∎Urine: - | ∎Blood: - ∎BM: + ∎Tissue:N ∎Urine: - |
| 7 | 32 / Male | 106 mo/History of *Cyclospora* infection | 7/ 180,000/ Yes (LPV/r + RAL) | Weight loss, diarrhea, fever, peripheral adenopathies, hepatomegaly, acute renal failure, anemia, ferritin elevation | Mediastinal, retroperitoneal adenopathies, hepatomegaly | No | Blood culture: negative BM culture: negative Bone biopsy pathology: *H. capsulatum* | Proven PDH/ Unspecified antifungal treatment/ Survived | ∎Blood: - ∎BM: - ∎Tissue: - ∎Urine: + | ∎Blood: + ∎BM: - ∎Tissue: NA ∎Urine: NA |

Abbreviation: mo = Months. AST = Aspartate aminotranferase. LDH = Lactic dehydrogenase. NUL = Normal upper limit. PCP = Pneumocystis pneumonia.

FTC = Emtricitabine. TDF = Tenofovir. EFV = Efavirenz. ABC = Abacavir. LPV/r = Lopinavir/ritonavir. RAL = Raltegravir. 3TC = Lamivudine. DRV = Darunavir.

RTV = Ritonavir. FLZ = Fluconazole. AMB = Amphotericin B. ITZ = Itraconazole. MAC = *Mycobacterium avium* Complex. BM = Bone marrow. NA = Not available.

In this study, we determined the diagnostic accuracy of the new generation antigen detection tests, the cHGEI IMMY, (monoclonal antibodies) and the MVHUALFA (polyclonal antibodies). The cHGEI IMMY showed good to excellent analytical performance, with an NPV of 96.6% in a population with a 35.9% prevalence of histoplasmosis, placing it as a good tool to rule out PDH in PLWHIV. These data agree with those of Cáceres et al. who found a sensitivity and specificity of 98% and 97% respectively, and NPV of 100% in 63 PLWHIV with PDH.[15] The improvement in diagnostic accuracy of the new test released by IMMY (cHGEI IMMY) is clear when compared to the previous version (IAHE). This test can be easily implemented at any clinical laboratory with medium infrastructure and personnel with moderate training or experience in performing ELISA tests. In addition, it is a rapid analysis that provides results in approximately 3 hours, it has CE [Conformité Européenne] Mark and FDA clearance. Furthermore, this test quantifies the concentration of antigen and leaves open the possibility of exploring its usefulness for the patient's follow-up, once the treatment has begun. One limitation of this test is the use of a standard curve of seven points, plus a positive and a negative control and a blank, then the laboratory has to collect several samples to optimize the use of the kit, which delays the diagnosis in settings where the test is requested unfrequently.

This is the first study to determine the performance of the MVHUALFA in urine samples to diagnose PDH in PLWHIV, with LR+ of 11.8 and LR- of 0.1. Cáceres et. al.[16] recently reported the performance of this test in serum of 75 PLWHIV with suspected histoplasmosis, 24 of them with proven-PDH. They found a sensitivity of 96% and specificity of 90%, by visual reading; while, with the use of an automated reader, the sensitivity and specificity were 92% and 94%, respectively. In our study, the result was read visually, with the presence of two bands, regardless of their intensity, as a positive result. In 20 samples, the observed result was

**Table 4. Clinical and laboratory characteristics of 25 patients with probable PDH and without additional diagnosis.**

| | Age/ Gender | Time since HIV diagnosis | Antiretro-viral treatment | Radiographic Pulmonary and extrapulmonary involvement | Empirical antifun-gal before samples | IAHE | cHGEI IMMY | MVHUA-LFA | Hcp100 nested PCR | 1281−1283₂₂₀ SCAR nested PCR |
|---|---|---|---|---|---|---|---|---|---|---|
| 1 | 39/Male | 39 mo | TDF + FTC + EFV | None | EQC (<72 h) | Positive | Positive | Positive | ▪Blood: - ▪BM: - ▪Tissue: NA ▪Urine: - | ▪Blood: + ▪BM: + ▪Tissue:NA ▪Urine: - |
| 2 | 40/Male | 2 mo | None | Micronodular infiltrates, hepatomegaly, splenomegaly | NA | Positive | Positive | Positive | ▪Blood: - ▪BM: - ▪Tissue: NA ▪Urine: NA | ▪Blood: + ▪BM: - ▪Tissue:NA ▪Urine: - |
| 3 | 52/Male | 0 mo | None | Mediastinal lymphadenopathies, hepatomegaly, splenomegaly | NA | Positive | Positive | Negative | ▪Blood: - ▪BM: - ▪Tissue: NA ▪Urine: NA | ▪Blood: - ▪BM: - ▪Tissue:NA ▪Urine: - |
| 4 | 60/Male | 2 mo | None | Micronodular and nodular infiltrates; neck and mediastinal lymphadenopathies | NA | Negative | Positive | Positive | ▪Blood: + ▪BM: + ▪Tissue: NA ▪Urine: + | ▪Blood: - ▪BM: + ▪Tissue:NA ▪Urine: - |
| 5 | 29/Male | 8 mo | ABC + 3TC LPV/r | Micronodular infiltrates, hepatomegaly, splenomegaly | >72h ITZ | Negative | Positive | Positive | ▪Blood: - ▪BM: NA ▪Tissue: NA ▪Urine: + | ▪Blood: - ▪BM: NA ▪Tissue:NA ▪Urine: - |
| 6 | 36/Male | 0 mo | None | Micronodular infiltrates | NA | Negative | Positive | Positive | ▪Blood: + ▪BM: - ▪Tissue: + ▪Urine: + | ▪Blood: - ▪BM: + ▪Tissue: - ▪Urine: - |
| 7 | 23/Male | 4 mo | TDF + FTC + DTG | Micronodular infiltrates, budd in tree infiltrates, neck and mediastinal lymphadenopathies, hepatomegaly | NA | Negative | Positive | Positive | ▪Blood: NA/NA ▪BM: - ▪Tissue: NA ▪Urine: - | ▪Blood: NA ▪BM: - ▪Tissue:NA ▪Urine: - |
| 8 | 36/Male | 107 mo | TDF + FTC + ABC + LPV/ r | NA | >72 h FLZ | Negative | Positive | Positive | ▪Blood: - ▪BM: - ▪Tissue: + ▪Urine: - | ▪Blood: + ▪BM: + ▪Tissue:NA ▪Urine: NA |
| 9 | 22/Male | 0 mo | None | Micronodular infiltrates, hepatomegaly | NA | Negative | Positive | Positive | ▪Blood: - ▪BM: - ▪Tissue: NA ▪Urine: NA | ▪Blood: + ▪BM: + ▪Tissue:NA ▪Urine: NA |
| 10 | 37/Male | 0 mo | TDF + FTC + EFV | Micronodular infiltrates | ITZ | Negative | Positive | Positive | ▪Blood: - ▪BM: - ▪Tissue: NA ▪Urine: - | ▪Blood: + ▪BM: + ▪Tissue:NA ▪Urine: NA |
| 11 | 64/Male | 0 mo | None | Consolidation infiltrates, budd in tree infiltrates, hepatomegaly | AMB | Negative | Positive | Negative | ▪Blood: - ▪BM: - ▪Tissue: NA ▪Urine: + | ▪Blood: + ▪BM: - ▪Tissue:NA ▪Urine: - |
| 12 | 21/Male | 0 mo | None | Micronodular infiltrates | FLZ | Negative | Positive | Negative | ▪Blood: - ▪BM: - ▪Tissue: NA ▪Urine: + | ▪Blood: - ▪BM: - ▪Tissue:NA ▪Urine: - |
| 13 | 21/ Female | 19 mo | TDF + FTC + EFV | Micronodular and consolidation infiltrates, hepatomegaly, splenomegaly | NA | Negative | Positive | Negative | ▪Blood: - ▪BM: - ▪Tissue: NA ▪Urine: NA | ▪Blood: - ▪BM: - ▪Tissue:NA ▪Urine: NA |
| 14 | 37/Male | 2 mo | None | NA | NA | Negative | Positive | Negative | ▪Blood: - ▪BM: - ▪Tissue: - ▪Urine: - | ▪Blood: + ▪BM: + ▪Tissue:NA ▪Urine: NA |

(*Continued*)

**Table 4.** (Continued)

| | Age/Gender | Time since HIV diagnosis | Antiretro-viral treatment | Radiographic Pulmonary and extrapulmonary involvement | Empirical antifun-gal before samples | IAHE | cHGEI IMMY | MVHUA-LFA | Hcp100 nested PCR | 1281−1283$_{220}$ SCAR nested PCR |
|---|---|---|---|---|---|---|---|---|---|---|
| 15 | 53/Male | 0 mo | None | Micronodular infiltrates, pleural effusion, neck, mediastinal and retroperitoneal lymphadenopathies, hepatomegaly | NA | Negative | Positive | Negative | ■Blood: -<br>■BM: -<br>■Tissue: NA<br>■Urine: NA | ■Blood: -<br>■BM: +<br>■Tissue:NA<br>■Urine: - |
| 16 | 41/Male | 174 mo | AZT + 3TC + EFV | Consolidation infiltrates, budd in tree infiltrates, hepatomegaly | NA | Negative | Positive | Negative | ■Blood: -<br>■BM: -<br>■Tissue: NA<br>■Urine: - | ■Blood: -<br>■BM: -<br>■Tissue:NA<br>■Urine: - |
| 17 | 40/Male | 1 mo | None | Micronodular infiltrates, pleural effusion, neck lymphadenopathies | NA | Negative | Positive | Negative | ■Blood: -<br>■BM: -<br>■Tissue: NA<br>■Urine: + | ■Blood: +<br>■BM: +<br>■Tissue:NA<br>■Urine: +/- |
| 18 | 53/Male | 0 mo | None | Micronodular and nodular infiltrates; hepatomegaly | FLZ | Positive | Negative | Negative | ■Blood: -<br>■BM: -<br>■Tissue: NA<br>■Urine: NA | ■Blood: -<br>■BM: -<br>■Tissue:NA<br>■Urine: - |
| 19 | 29/Female | 106 mo | 3TC + AZT + LPV/r | Neck and mediastinal lymphadenopathies, hepatomegaly, splenomegaly | NA | Positive | Negative | Negative | ■Blood: -<br>■BM: -<br>■Tissue: NA<br>■Urine: NA | ■Blood: -<br>■BM: -<br>■Tissue:NA<br>■Urine: + |
| 20 | 43/Male | 190 mo | ABC + 3TC + AZT + ATV/r | Micronodular infiltrates, budd in tree, cavitary lesions, retroperitoneal lymphadenopathies, hepatomegaly | ITZ | Positive | Negative | Negative | ■Blood: -<br>■BM: -<br>■Tissue: NA<br>■Urine: NA | ■Blood: +<br>■BM: +<br>■Tissue:NA<br>■Urine: + |
| 21 | 38/Male | 0 mo | None | Abdominal and retroperitoneal lymphadenopathies, hepatomegaly | NA | Positive | Negative | Negative | ■Blood: NA<br>■BM: -<br>■Tissue: NA<br>■Urine: - | ■Blood: NA<br>■BM: -<br>■Tissue:NA<br>■Urine: NA |
| 22 | 41/Male | 2 mo | TDF + FTC + EFV | Micronodular infiltrates, hepatomegaly, | AMB | Negative | Negative | Positive | ■Blood: -<br>■BM: -<br>■Tissue: NA<br>■Urine: - | ■Blood: -<br>■BM: -<br>■Tissue:NA<br>■Urine: - |
| 23 | 45/Male | 0 mo | None | NA | NA | Negative | Negative | Positive | ■Blood: -<br>■BM: -<br>■Tissue: NA<br>■Urine: - | ■Blood: +<br>■BM: +<br>■Tissue:NA<br>■Urine: NA |
| 24 | 35/Female | 0 mo | None | Consolidation infiltrates, intestinal lesions | FLZ | Negative | Negative | Positive | ■Blood: -<br>■BM: -<br>■Tissue: NA<br>■Urine: - | ■Blood: +<br>■BM: +<br>■Tissue:NA<br>■Urine: - |
| 25 | 30/Male | 8 mo | TDF + FTC + RAL | Consolidation and nodular infiltrates, pleural effusion; neck, axillar, retroperitoneal, inguinal lymphadenopathies, hepatomegaly, splenomegaly | NA | Negative | Negative | Positive | ■Blood: +<br>■BM: +<br>■Tissue: NA<br>■Urine: NA | ■Blood: -<br>■BM: -<br>■Tissue:NA<br>■Urine: + |

Abbreviations: IAHE = IMMY ALPHA *Histoplasma* EIA kit. cHGEI IMMY = clarus *Histoplasma* GM Enzyme Immunoassay. MVHUALFA = MiraVista *Histoplasma* Urine Antigen LFA. Mo = Months, TDF = Tenofovir, FTC = Emtricitabine, EFV = Efavirenz, ABC = Abacavir, 3TC = Lamivudine, LPV/r = Lopinavir/ritonavir, DTG = Dolutegravir, AZT = Zidovudine, ATV/r = Atazanavir/ritonavir, RAL = Raltegravir, EQC = Echinocandins. ITZ = Itraconazole. FLZ = Fluconazole. AMB = Amphotericin B. BM = Bone marrow, NA = Not available.

**Table 5. Clinical and laboratory characteristics of 16 patients with probable PDH and with an additional diagnosis.**

| | Age / Gender | Time since HIV diagnosis | Antiretroviral treatment | Radiographic Pulmonary and extrapulmonary involvement | Empirical antifungal before samples | Final diagnosis | IAHE | cHGEI IMMY | MVHUA-LFA | Hcp100 nested PCR | 1281−1283_{220} SCAR nested PCR |
|---|---|---|---|---|---|---|---|---|---|---|---|
| 1 | 28/Male | 0 mo | None | Micronodular infiltrates | <72 h AMB | *C. neoformans* | Positive | Positive | Positive | ■Blood: +<br>■BM: +<br>■Tissue: NA<br>■Urine: + | ■Blood: +<br>■BM: +<br>■Tissue: NA<br>■Urine: + |
| 2 | 33/Male | 80 mo | TDF + AZT + LPV/r | Micronodular infiltrates, budd in tree, mediastinal lymphadenopathies, hepatomegaly, splenomegaly | NA | *M. tuberculosis* | Positive | Positive | Negative | ■Blood: +<br>■BM: +<br>■Tissue: NA<br>■Urine: NA | ■Blood: +<br>■BM: +<br>■Tissue: NA<br>■Urine: - |
| 3 | 53/Male | 0 mo | None | Hepatomegaly and splenomegaly | ITZ and AMB | *Malbranchea* spp. | Negative | Positive | Positive | ■Blood: -<br>■BM: +<br>■Tissue: +<br>■Urine: NA | ■Blood: +<br>■BM: -<br>■Tissue: -<br>■Urine: - |
| 4 | 33/Female | 2 mo | None | NA | FLZ and ITZ | Granulomatous hepatitis | Negative | Positive | Positive | ■Blood: +<br>■BM: -<br>■Tissue: -<br>■Urine: + | ■Blood: -<br>■BM: +<br>■Tissue: +<br>■Urine: + |
| 5 | 33/Male | 0 mo | None | Consolidation and nodular infiltrates, pleural effusion, abdominal, inguinal and retroperitoneal adenopathies, hepatomegaly, splenomegaly, pericardial effusion | FLZ | *M. bovis*, CMV colitis, HPV infection | Negative | Positive | Negative | ■Blood: -<br>■BM: NA<br>■Tissue: NA<br>■Urine: - | ■Blood: -<br>■BM: NA<br>■Tissue: NA<br>■Urine: - |
| 6 | 27/Male | 2 mo | 3TC +ABC +EFV | Pleural effusion, mediastinal effusion, neck, mediastinal lymphadenopathies, hepatomegaly, splenomegaly | NA | *M. tuberculosis* | Negative | Positive | Negative | ■Blood: +<br>■BM: -<br>■Tissue: +<br>■Urine: + | ■Blood: +<br>■BM: +<br>■Tissue: +<br>■Urine: + |
| 7 | 28/Male | 0 mo | None | Micronodular infiltrates, axillar and mediastinal lymphadenopathies, hepatomegaly, splenomegaly | NA | *M. tuberculosis* | Negative | Positive | Negative | ■Blood: -<br>■BM: -<br>■Tissue: NA<br>■Urine: + | ■Blood: -<br>■BM: +<br>■Tissue: NA<br>■Urine: - |
| 8 | 27/Male | 82 mo | LPV/r + TDF + FTC + EFV | Cavitary lesions | NA | *C. neoformans* | Positive | Negative | Negative | ■Blood: -<br>■BM: -<br>■Tissue: NA<br>■Urine: NA | ■Blood: -<br>■BM: -<br>■Tissue: NA<br>■Urine: NA |
| 9 | 46/Male | 0 mo | None | Consolidations and micronodular infiltrates | ITZ | *C. neoformans* | Positive | Negative | Negative | ■Blood: +<br>■BM: +<br>■Tissue: NA<br>■Urine: NA | ■Blood: +<br>■BM: +<br>■Tissue: NA<br>■Urine: - |
| 10 | 24/Male | 3 mo | None | Micronodular infiltrates | FLZ | *M. avium* | Negative | Negative | Positive | ■Blood: +<br>■BM: +<br>■Tissue: NA<br>■Urine: - | ■Blood: -<br>■BM: -<br>■Tissue: NA<br>■Urine: - |

*(Continued)*

**Table 5.** (Continued)

| | Age / Gender | Time since HIV diagnosis | Antiretroviral treatment | Radiographic Pulmonary and extrapulmonary involvement | Empirical antifungal before samples | Final diagnosis | IAHE | cHGEI IMMY | MVHUA-LFA | Hcp100 nested PCR | 1281 −1283₂₂₀ SCAR nested PCR |
|---|---|---|---|---|---|---|---|---|---|---|---|
| 11 | 38/Male | 1 mo | None | Consolidations and micronodular infiltrates; neck, mediastinal, inguinal lymphadenopathies | NA | HIV associated lymphadenopathy | Negative | Negative | Positive | ∎Blood: - ∎BM: - ∎Tissue: NA ∎Urine: - | ∎Blood: + ∎BM: + ∎Tissue: + ∎Urine: - |
| 12 | 48/Male | 132 mo | TDF + FTC + EFV | Nodular infiltrates, pleural effusion; neck, axillar, mediastinal lymphadenopathies; CNS lesions | FLZ and AMB | CMV proctitis | Negative | Negative | Positive | ∎Blood: - ∎BM:NA ∎Tissue: - ∎Urine: NA | ∎Blood: - ∎BM:NA ∎Tissue: - ∎Urine: - |
| 13 | 36/Male | 138 mo | None | Micronodular infiltrates; mediastinal, axillar, abdominal, retroperitoneal, inguinal lymphadenopathies; hepatomegaly, splenomegaly | NA | *M. bovis* | Negative | Negative | Positive | ∎Blood: + ∎BM: + ∎Tissue: - ∎Urine: - | ∎Blood: + ∎BM: - ∎Tissue: NA ∎Urine: + |
| 14 | 45/Male | 2 mo | TDF + FTC + EFV | Nodular infiltrates, pleural effusion; cervical, mediastinal, abdominal, retroperitoneal, inguinal lymphadenopathies; hepatomegaly, splenomegaly | NA | *M. avium* and Sarcoma Kaposi | Negative | Negative | Positive | ∎Blood: - ∎BM: + ∎Tissue: + ∎Urine: NA | ∎Blood: - ∎BM: + ∎Tissue: NA ∎Urine: - |
| 15 | 36/Male | 6 mo | TDF + FTC + EFV | Nodular lung infiltrates, neck, mediastinal, abdominal, inguinal and retroperitoneal lymphadenopathies; hepatomegaly, splenomegaly and pericardial effusion | AMB | *M. avium* | Negative | Positive | Positive | ∎Blood: - ∎BM: - ∎Tissue: - ∎Urine: - | ∎Blood: + ∎BM: - ∎Tissue: + ∎Urine: - |
| 16 | 21/Male | 0 mo | TDF + FTC + EFV | Mediastinal and retroperitoneal lymphadenopathies, gastrointestinal lesions | NA | *M. tuberculosis* | Negative | Positive | Negative | ∎Blood: - ∎BM: - ∎Tissue: NA ∎Urine: NA | ∎Blood: - ∎BM: - ∎Tissue: NA ∎Urine: - |

Abbreviation: IAHE = IMMY ALPHA *Histoplasma* EIA kit. cHGEI IMMY = clarus *Histoplasma* GM Enzyme Immunoassay. MVHUALFA = MiraVista *Histoplasma* Urine Antigen LFA. mo = Months. TDF = Tenofovir. AZT = Zidovudine. LPV/r = Lopinavir/ritonavir. 3TC = Lamivudine. ABC = Abacavir. EFV = Efavirenz. FTC = Emtricitabine. AMB = Amphotericin B. ITZ = Itraconazole. FLZ = Fluconazole. BM = Bone marrow. NA = Not available.

weakly positive, 14 were cases of proven-PDH, and six without proven-PDH according to the gold standard; however, one of them had a positive result by both tests, cHGEI IMMY and MVHUALFA. Definitely, the use of the automated reader should be considered in order to avoid reading errors, although this test has the great advantage of being presented in a single format, less than a minute is required to perform the test, and 40 minutes to read the result, which allows it to really work as a point of care test.

Either test is an excellent choice, both to rule out or to confirm PDH in PLWHIV. The integration of these tests in clinical laboratories will certainly impact on early diagnosis and treatment, and therefore on the outcome of patients. The selection will surely depend on its availability in different countries, on the infrastructure available in the laboratories, existence of trained personnel in carrying out the tests, and of course, on the costs.

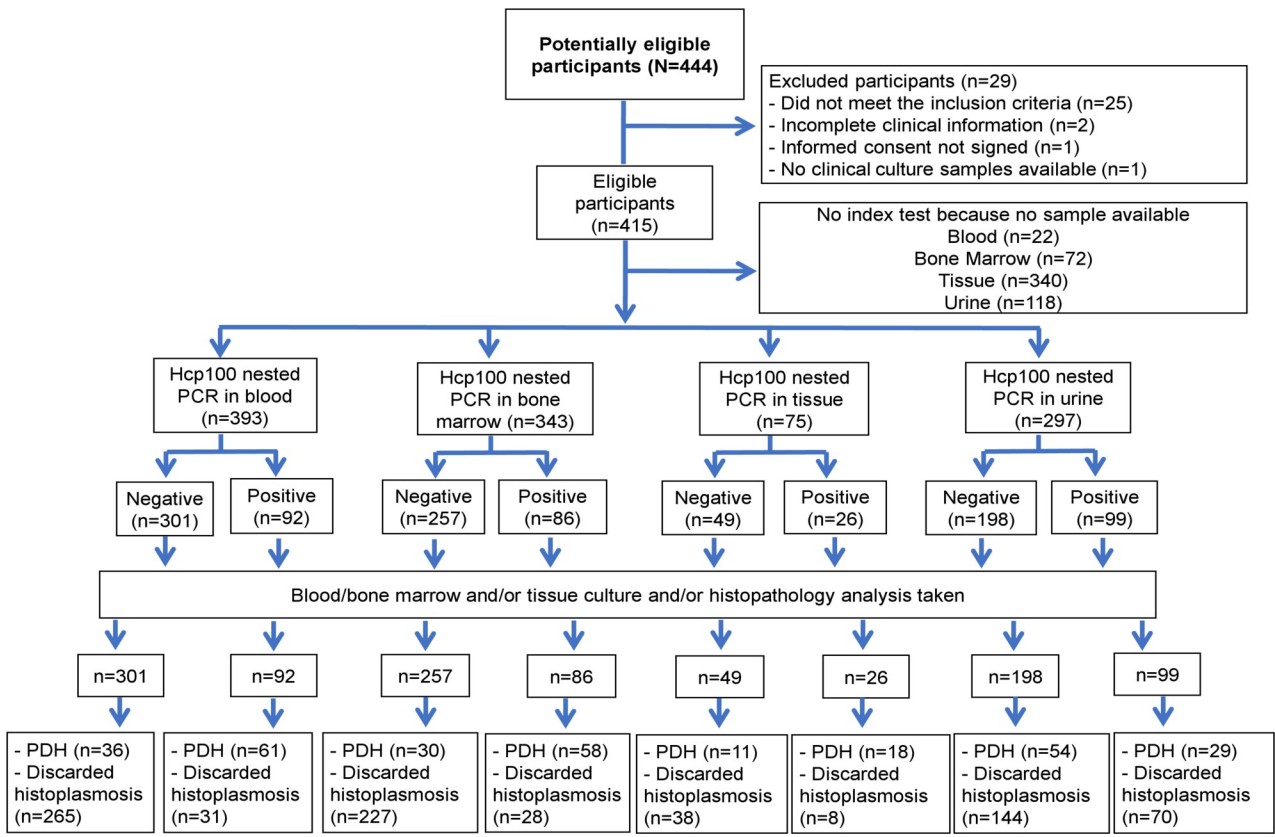

**Fig 4. Flowchart of the samples subjected to Hcp100 nested PCR.** Abbreviation: PDH = Progressive disseminated histoplasmosis. No inconclusive results were found.

Based on several limitations of the traditional gold standard tests as are those related to laboratory infrastructure, staff training, analytical performance, turnaround time for diagnosis; and the benefits of the antigen tests (high diagnosis accuracy, kits commercially available, lower laboratory biosecurity level, ease performance) the 2020 PAHO/WHO Guidelines for the diagnosis and managing disseminated histoplasmosis among PLWHIV recommends "disseminated histoplasmosis should be diagnosed among PLWHIV by detecting circulating *Histoplasma* antigens", it is acknowledge that financial, technical and implementation strategies are needed to achieved this goal. [26,27] In the same way, one of the objectives of the International Histoplasmosis Advocacy Group is that at least one laboratory in all of the countries in the Americas have rapid and accurate tests available for the diagnosis of histoplasmosis.[28]

An important issue to keep in mind, especially in endemic areas, is that these tests have shown cross-reactivity with *Blastomyces*, *Coccidiodes*, *Paracoccidioides* [13,14]. Thus, of the 41 patients with probable-PDH, in 25 patients no other diagnosis was defined and among the other 16 patients different co-infections and inflammatory conditions were diagnosed. In this report, we found three patients with probable-PDH and *C. neoforman*s infection (isolated both in blood and in bone marrow cultures), and one case of proven-PDH with *C. neoformans* recovered from blood. Cryptococcosis and histoplasmosis are common fungal infections in PLWHIV, however, *Cryptococcus*/*Histoplasma* co-infection is rare, there are only a few reports in patients with advanced HIV and CD4 lymphocyte count <100 cells / mm3, likewise, there is a report of a case without HIV, with chronic treatment with steroids. This co-infection is likely underestimated due to the faster growth of *C. neoformans* and the slower growth rate of

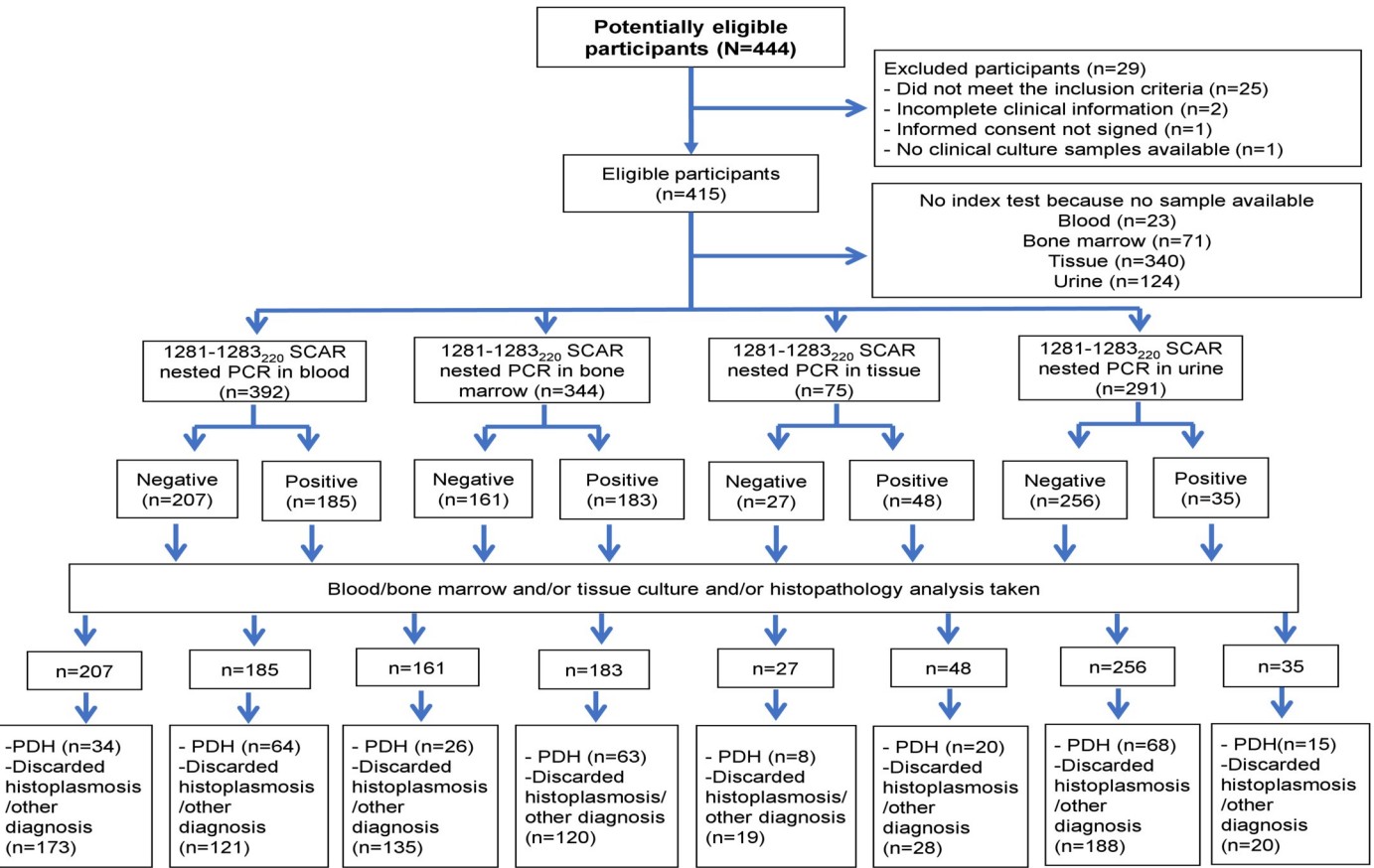

**Fig 5. Flowchart of the samples subjected to 1281−1283$_{220}$ SCAR nested PCR.** Abbreviation: PDH = Progressive disseminated histoplasmosis. No inconclusive results were found.

*Histoplasma*. Likewise, the diagnosis of one of them can cause the underestimation of the existence of the other.[29,30,31]

Among the patients with probable-PDH without other added diagnosis, 8/25 had a positive Hcp100 nested PCR result and 16/25 a positive 1281−1283$_{220}$ nested PCR result. Meanwhile, among the patients with other co-infections 10/16 had a Hcp100 nested PCR positive result or a positive 1281−1283$_{220}$ SCAR nested PCR result. An interesting finding that opens the possibility of using molecular tests as diagnostic support, although they showed variable data on sensitivity and specificity.

Seven proven-PDH cases showed false negative results in the three antigen detection tests. In two previous studies, false negative results have been attributed to the probable low concentration of circulating antigen as a result of using TMP/SMX as a treatment for other infections in PLWHIV,[15,16] since this antibiotic has shown *in−vitro* activity against *Histoplasma*.[32] Although, we do not have the information on the use of TMP/SMX, it is commonly used in PLWHIV as prophylaxis against *Pneumocystis jirovecii*. Another possibility that has not been explored, is the interference of foods, vaginal creams, consumption of caffeine, acetaminophen and acetylsalicylic acid is not known yet. Of these patients, 2/7 had positive cultures and 5/7 showed positive histopathological results. The Hcp 100 nested PCR showed positivity in 3/7, while 1281−1283$_{220}$ SCAR nested PCR was positive in 6/7. Although the sensitivity of the PCR tests was more or less similar, the specificity of the nested Hcp100 PCR (between 892.6% and

89.5%) in blood, bone marrow and tissue samples was higher than the specificity observed with the SCAR 1281 PCR -$1283_{220}$ (between 40.4% and 58.8%).

One of the most widely used molecular tools for the detection of *Histoplasma* has been the Hcp100 gene. The data reported on the sensitivity and specificity of this PCR test are greater than 90%. However, the majority of these studies are quite different in terms of population studied, samples analyzed, extraction protocols, PCR platforms, use of DNAs from *Histoplasma* isolate collections, as well as, inclusion of healthy patients to determine the specificity of the test.[17] Moreover, there are only a few studies in PLWHIV and suspected PDH, where the investigators reported sensitivity from 89 to 100% and specificity from 98 to 100% using different fluids.[33–36] In this multicenter comparative trial the sensitivity observed was from 62.1 to 65.9% in blood, bone marrow or tissue biopsies and the specificity from 82.6 to 89.5%. The diagnostic accuracy of Hcp100 nested PCR in urine, which may be the sample of choice for diagnosis, due to the ease to obtain it, showed the lowest analytical performance, with 34% sensitivity and 67% specificity. The performance of the 1281–1283220 SCAR nested PCR was generally poor, had a LR+ of 1.2 to 1.9 and LR- of 0.55 to 0.9, likewise, the Hcp100 nested PCR only showed a performance between regular and good, even when performed on bone marrow samples. Even though, this test has shown some utility in studying histoplasmosis in occupational and recreational outbreaks.[37,19]

In the study of Frias et al. 2011,[18] the specificity of the 1281–$1283_{220}$ SCAR markers was determined by performing PCR from genomic DNA. In this study the description of the method and its initial analytical validation were carried out. In our multicenter, prospective, double-blind cohort study of patients with HIV and suspected diagnosis of disseminated and progressive histoplasmosis, we are describing the real diagnostic performance of this molecular test. This may explain the differences in specificity found in both studies.

Two major disadvantages towards the use of nucleic acid amplification techniques are the lack of standardized protocols given that there is still no consensus on which is the best sample, number of samples to analyze, DNA extraction protocol, target or platform for amplification, and trained personnel are needed to perform these types of tests.

Our study has some limitations, we did not evaluate the possible interference of foods, creams, medications that could influence the performance of the tests, nor we considered the intentional search for infection by blastomycosis, coccidioidomycosis and paracoccidioidomycosis, the main agents that show cross-reaction with the *Histoplasma* antigen detection tests. [11,14]

We consider that our study has the strength to determine the performance of rapid tests for the diagnosis of histoplasmosis—antigen detection in urine and molecular tests—which are the two modalities proposed to make them available in at least one city in each country of America.[28] Furthermore, the performance of these tests was determined by comparing them with the gold standard (culture/histopathology). Additionally, we started from the clinical definition of suspected cases of PDH that reflects the daily clinical reasoning that triggers the diagnostic approach and allowed us to evaluate each test in a real-life setting.

In conclusion, urine *Histoplasma* antigen detection tests showed excellent performance for the diagnosis of PDH in PLWHIV. The integration of these tests in clinical laboratories will certainly impact on early diagnosis and treatment, and consequently on the outcome of patients. These tests have an advantage over PCR tests, their functioning is higher, they are commercial standardized tests, easy and quick to perform, and provide results in minutes or hours. Therefore, the integration of these tests in clinical laboratories will certainly impact on early diagnosis/treatment and consequently on the outcome of the patients.

## Supporting information

**S1 STARD Checklist. Standards for Reporting Diagnostic accuracy studies.**
(PDF)

## Acknowledgments

The authors thank the personnel of the Laboratorio Nacional de Máxima Seguridad para el Estudio de la Tuberculosis y Enfermedades Emergentes, CONACYT/Instituto Nacional de Ciencias Médicas y Nutrición Salvador Zubirán for their invaluable help with sample´s processing.

We thank IMMY and Dr. L. Joseph Wheat (MiraVista Diagnostic Laboratories) for providing us with the necessary kits to perform this study. Neither of them influenced the study design, analysis and interpretation of data, writing and publication of results.

## Author Contributions

**Conceptualization:** Areli Martínez-Gamboa, María Dolores Niembro-Ortega, Pedro Torres-González, Brenda Crabtree-Ramírez, Pedro Martínez-Ayala, Patricia Rodríguez-Zulueta, Alfredo Ponce de León-Garduño, José Sifuentes-Osornio.

**Data curation:** Areli Martínez-Gamboa, María Dolores Niembro-Ortega, Edgardo Reyes-Gutiérrez, Víctor Hugo Ahumada-Topete, Jaime Andrade-Villanueva, Antonio Olivas-Martínez.

**Formal analysis:** Areli Martínez-Gamboa, María Dolores Niembro-Ortega, Pedro Torres-González, Antonio Olivas-Martínez.

**Funding acquisition:** Areli Martínez-Gamboa.

**Investigation:** Areli Martínez-Gamboa, María Dolores Niembro-Ortega, Pedro Torres-González, Armando Gamboa-Domínguez, Gustavo Reyes-Terán, Víctor Hugo Lozano-Fernandez, Víctor Hugo Ahumada-Topete, Pedro Martínez-Ayala, Marisol Manríquez-Reyes, Juan Pablo Ramírez-Hinojosa, Christian Hernández-León, Jesús Ruíz-Quiñones, Norma Eréndira Rivera-Martínez, Alberto Chaparro-Sánchez, Luz Alicia González-Hernández, Sofia Cruz-Martínez, Oscar Flores-Barrientos, Jesús Enrique Gaytán-Martínez, Martín Magaña-Aquino, Javier Araujo-Meléndez, María del Rocío Reyes-Montes, María Lucia Taylor.

**Methodology:** Areli Martínez-Gamboa, María Dolores Niembro-Ortega, Pedro Torres-González, Janeth Santiago-Cruz, Nancy Guadalupe Velázquez-Zavala, Andrea Rangel-Cordero, Jaime Andrade-Villanueva, Axel Cervantes-Sánchez, Esperanza Duarte-Escalante, María Guadalupe Frías-De León, José Antonio Ramírez.

**Supervision:** Alfredo Ponce de León-Garduño, José Sifuentes-Osornio.

**Writing – original draft:** Areli Martínez-Gamboa, María Dolores Niembro-Ortega, Nancy Guadalupe Velázquez-Zavala, Brenda Crabtree-Ramírez, Armando Gamboa-Domínguez, Pedro Martínez-Ayala, Juan Pablo Ramírez-Hinojosa, Christian Hernández-León, Alberto Chaparro-Sánchez, Sofia Cruz-Martínez.

**Writing – review & editing:** Janeth Santiago-Cruz, Andrea Rangel-Cordero, Edgardo Reyes-Gutiérrez, Gustavo Reyes-Terán, Víctor Hugo Lozano-Fernandez, Víctor Hugo Ahumada-Topete, Marisol Manríquez-Reyes, Patricia Rodríguez-Zulueta, Jesús Ruíz-Quiñones, Norma Eréndira Rivera-Martínez, Jaime Andrade-Villanueva, Luz Alicia González-

Hernández, Oscar Flores-Barrientos, Jesús Enrique Gaytán-Martínez, Martín Magaña-Aquino, Antonio Olivas-Martínez, Javier Araujo-Meléndez, María del Rocío Reyes-Montes, Esperanza Duarte-Escalante, María Guadalupe Frías-De León, José Antonio Ramírez, María Lucia Taylor, Alfredo Ponce de León-Garduño, José Sifuentes-Osornio.

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
