## [Decision Letter · Decision Letter 0]

13 Nov 2020

Dear Dr. Sifuentes-Osornio,

Thank you very much for submitting your manuscript "Diagnostic accuracy of antigen detection in urine and molecular assays testing in different clinical samples for the diagnosis of progressive disseminated histoplasmosis in patients living with HIV/AIDS: A prospective multicenter study in Mexico." for consideration at PLOS Neglected Tropical Diseases. As with all papers reviewed by the journal, your manuscript was reviewed by members of the editorial board and by several independent reviewers. In light of the reviews (below this email), we would like to invite the resubmission of a significantly-revised version that takes into account the reviewers' comments. 

We cannot make any decision about publication until we have seen the revised manuscript and your response to the reviewers' comments. Your revised manuscript is also likely to be sent to reviewers for further evaluation.

Sincerely,

Angel Gonzalez, Ph.D.

Associate Editor

Todd Reynolds

Deputy Editor

Reviewer's Responses to Questions

**Key Review Criteria Required for Acceptance?**

**Methods**

-Are the objectives of the study clearly articulated with a clear testable hypothesis stated?

-Is the study design appropriate to address the stated objectives?

-Is the population clearly described and appropriate for the hypothesis being tested?

-Is the sample size sufficient to ensure adequate power to address the hypothesis being tested?

-Were correct statistical analysis used to support conclusions?

-Are there concerns about ethical or regulatory requirements being met?

Reviewer #1: Methods

 -Are the objectives of the study clearly articulated with a clear testable hypothesis stated? Yes

 -Is the study design appropriate to address the stated objectives? Yes

 -Is the population clearly described and appropriate for the hypothesis being tested? Yes

 -Is the sample size sufficient to ensure adequate power to address the hypothesis being tested? Yes

 -Were correct statistical analysis used to support conclusions? Yes

 -Are there concerns about ethical or regulatory requirements being met? Yes

Comments about methods 

1. Methods: please in “clinical definitions” add the EORCT/MSGREC reference. 

2. Methods: please in “antigen detection procedures in urine samples” add more details of kits used (please see first major comment). 

3. Methods: please check in “Histoplasma antigen detection in urine, using the IMMY ASR GM EIA” please check line 203 “The optical densities (OD) were read at 450 nm”. Product package insert said on reading the test: “A dual wavelength reader is required, with absorbances read at 450 nm and 620/630 nm. Blank on the 1X Wash Buffer ( ). This assay has not been validated with a single wavelength reader”. Additional, please correct in line 202; “0·4” by “0.4”.

Reviewer #2: Authors provided us with an important study on the comparaison of several diagnostic tools using similar samples collected prospectively and consecutively.

Objectives, case-definition and populations are clearly explained.

The methods section informed us on the samples origin, management and detailed procedures on how they perform analyses.

Statistics are correct and adapted to the design. Still no information regarding the statistical power was given. But this can be approach using the confidence interval mentionned when necessary accross the manuscript.

Ethics met probably the mexican regulations but not sure since the study was not registered at the national level.

**Results**

-Does the analysis presented match the analysis plan?

-Are the results clearly and completely presented?

-Are the figures (Tables, Images) of sufficient quality for clarity?

Reviewer #1: -Does the analysis presented match the analysis plan? Yes

 -Are the results clearly and completely presented? Yes

 -Are the figures (Tables, Images) of sufficient quality for clarity? No, please improve resolution

Comments about results

1. Results: this a great multicenter study, that add valuable information about histoplasmosis in Mexico (10 study places from seven Mexico states). I would like to suggest, add a map describing patient tested by region, and positivity by region. Please check figure 3 of this study published by Falci et al in Brazil (https://doi.org/10.1093/ofid/ofz073). 

2. Results: authors did a great description of the 280 patients without histoplasmosis, in especial the description of co-infections. Is it same information available for the histoplasmosis cases? It is available, please added it. I am very curious in special for mycobacterial co-infections. 

3. Results: Please, in table 3 add value of assay accuracy (this value is important to know correct proportion of classification).

4. Results: Line 411 and 424, please clarified that this false negative and positive results are from the antigen results. 

5. Results: “false positive” On these 26 patients is available data of serology (immunodiffusion or complement Fixation)? What happened with data from molecular testing on these 26 patients? I think data from Hcp 100 nested PCR would be useful (not too sensitive, but specific). 

6. Please improve figures resolution, I was not able to see information presented there (figures pending to review).

Reviewer #2: Flow charts are comprehensive and in compliance with the written results section.

Results matched the plan analyses.

Results are clearly and completely presented with results in appendix to help let the reader consult the overall detailed results.

There are several tables and figures. If all of them are useful it may be helpful to merge tables as recommended below.

**Conclusions**

-Are the conclusions supported by the data presented?

-Are the limitations of analysis clearly described?

-Do the authors discuss how these data can be helpful to advance our understanding of the topic under study?

-Is public health relevance addressed?

Reviewer #1: -Are the conclusions supported by the data presented? Yes

 -Are the limitations of analysis clearly described? Yes

 -Do the authors discuss how these data can be helpful to advance our understanding of the topic under study? Yes

 -Is public health relevance addressed? Yes

Comments about

1. Discussion: Analytical performance of molecular test was not the best, but I consider these negative results add valuable information. Reading the discussion, I got the felling this data was ignored at the end of the manuscript. Please include some points like the detection of DNA in six of the seven false negatives of the antigen test. And also discuss the specificity issue with the 1281-1283220 SCAR nested PCR (compared with original validations results, Frias et at 2011).

Reviewer #2: Conclusions and limitations are well expressed with considerations to the lack of informations regarding the main fungal infections that have described to date due to due to the study design.

Authors expressed on how there resuls are important in the field with considerations to he public health impact of different patients' management strategies.

**Editorial and Data Presentation Modifications?**

Reviewer #1: Diagnostic accuracy of antigen detection in urine and molecular assays testing in different clinical samples for the diagnosis of progressive disseminated histoplasmosis in patients living with HIV/AIDS: A prospective multicenter study in Mexico.

This is a well prepared manuscript. Data presented in this manuscript is valuable and will add important information for the improvement of histoplasmosis diagnosis and care. 

I would like to highlight the good design of this study, and the way data was presented. This study evaluated a large number of patients, and lot of laboratory and clinical data. Information presented is clear and easy to follow and understand (congratulation to authors).

I think data presented here about molecular testing is also valuable.

Some aspect needs to be improved, in order to do this manuscript much better.

Minor:

1. Please check the use of italics in microorganism names, in special “Histoplasma”. 

2. Did you have any indeterminate results using the MiraVista LFA?

3. Do you have data of the Histoplasma EIA concentration? Can you present the median and rank of the concentration?

4. On Tables 1 and 2: it is possible to add data about co-infections and add CD4<100? Please also correct and extra comma after some of the values (age, HIV diagnosis, CD4, HIV VL…). Addtionally, in signs and symptoms, and X rays, I would like to suggest listed these from more frequent to less frequent.

5. Table 1: correct proportion of diarrhea (correct is 94.9%, or you can use 95.0%)

Major:

1. Please mention antigen test commercial names and products references. Please check products web-site. 

a. Clarus HISTOPLASMA GM ENZYME IMMUNOASSAY (https://www.immy.com/hgm) 

b. MiraVista Histoplasma urine antigen LFA (https://www.immy.com/hgm)

2. The IMMY ALPHA Histoplasma EIA is a product discontinued, this product was replaced by the Clarus HISTOPLASMA GM ENZYME IMMUNOASSAY (the IMMY ASR GM EIA your evaluated here). Do you think it is necessary to present this data? In special based in the fact that this data was previous published? (PLOS NTD Nov 5, 2018). I would like to suggest removed it, this could be help on shorter the final paper, and also help on avoid confusion on people not experts on histoplasmosis diagnosis. 

3. Introduction: Please add more details in the introduction about the problem of the histoplasmosis and tuberculosis co-infection. Here some recommended references:

a. Adenis et al. Am J Trop Med Hyg. 2014 Feb;90(2):216-23.

b. Nacher et al. PLoS Negl Trop Dis. 2014 Dec 4;8(12):e3290.

c. Adenis et al. Lancet Infect Dis. 2018 Oct;18(10):1150-1159.

d. Caceres and Valdes. J Fungi (Basel). 2019 Aug 9;5(3):73.

4. Introduction: Please also mention in the introduction that the IMMY ASR GM EIA and the MiraVista lateral flow assay were previous validated, here the references:

a. Caceres et al. Mycoses. 2020 Feb;63(2):139-144.

b. Caceres et al. J Clin Microbiol. 2018 May 25;56(6):e01959-17.

5. Methods: please in “clinical definitions” add the EORCT/MSGREC reference. 

6. Methods: please in “antigen detection procedures in urine samples” add more details of kits used (please see first major comment). 

7. Methods: please check in “Histoplasma antigen detection in urine, using the IMMY ASR GM EIA” please check line 203 “The optical densities (OD) were read at 450 nm”. Product package insert said on reading the test: “A dual wavelength reader is required, with absorbances read at 450 nm and 620/630 nm. Blank on the 1X Wash Buffer ( ). This assay has not been validated with a single wavelength reader”. Additional, please correct in line 202; “0·4” by “0.4”. 

8. Results: this a great multicenter study, that add valuable information about histoplasmosis in Mexico (10 study places from seven Mexico states). I would like to suggest, add a map describing patient tested by region, and positivity by region. Please check figure 3 of this study published by Falci et al in Brazil (https://doi.org/10.1093/ofid/ofz073). 

9. Results: authors did a great description of the 280 patients without histoplasmosis, in especial the description of co-infections. Is it same information available for the histoplasmosis cases? It is available, please added it. I am very curious in special for mycobacterial co-infections. 

10. Results: Please, in table 3 add value of assay accuracy (this value is important to know correct proportion of classification).

11. Results: Line 411 and 424, please clarified that this false negative and positive results are from the antigen results. 

12. Results: “false positive” On these 26 patients is available data of serology (immunodiffusion or complement Fixation)? What happened with data from molecular testing on these 26 patients? I think data from Hcp 100 nested PCR would be useful (not too sensitive, but specific). 

13. Please improve figures resolution, I was not able to see information presented there (figures pending to review).

14. Discussion: Analytical performance of molecular test was not the best, but I consider these negative results add valuable information. Reading the discussion, I got the felling this data was ignored at the end of the manuscript. Please include some points like the detection of DNA in six of the seven false negatives of the antigen test. And also discuss the specificity issue with the 1281-1283220 SCAR nested PCR (compared with original validations results, Frias et at 2011). 

As final comments, I hope the author with data of this study will prepare a second manuscript comparing patients with and without histoplasmosis.

Reviewer #2: Minor revision.

METHODS

Ag detection procedures, sample preparation of molecular assays and molecular assay procedures could be pushed to an appendix to commit to the objective of reducing the word number

Regarding Ag detection procedures, information regarding the number of wells per patients is missing and also what was the threshold considered regaring CV in order to decide if a new test is necessary or not.

Are there samples that were run several times for any reasons or not? This could give informations regarding quality control procedures performed or not on the different run.

RESULTS

Authors should merge Table 1 and 2 to save the number of tables and give a comprehensive overview of baseline data for the total population, the proven Progressive Disseminated Histoplasmosis (PDH) cases and add both the characteristics of probable PDH and patients without proven PDH.

First column with the 415 patients, 2nd clumn with the 108 confirmed, 27 probable and 280 without PDH, raw data column results should express n/N and percentage in (). 

This table will enable harmonization of the raw data because informations was missed in between the two tables.

Please correct typos in the tables, notably kommas duplicate or missing informations after kommas.

If available in population characteristics, time from ART initiation would be of high relevance since about 1/2 patients included and similar for PDH proven are not ART naive, informing on the context of potential infectious IRIS when discovering/diagnosing the disease. The context of IRIS may be relevant when discussing the false positive and negative patients.

As the information is probably available, case-fatality rate should be expressed in the four populations listed above.

It is not clearly explain why the total number and the number of PDH-proven urine samples for antigen detection varies from one test to another.

Please clarify. It may help explain variation in agreement in between antigen detection tests.

Another proposal, which may have a low impact on statistical power would be to only consider those samples that were available for all antigen detection methods.

DISCUSSION

An important reference is lacking. Authors should have mentionned the recently release WHO Guidelines on diagnosing and treating histoplasmosis in people living with HIV.

**Summary and General Comments**

Reviewer #1: This is a well prepared manuscript. Data presented in this manuscript is valuable and will add important information for the improvement of histoplasmosis diagnosis and care. 

I would like to highlight the good design of this study, and the way data was presented. This study evaluated a large number of patients, and lot of laboratory and clinical data. Information presented is clear and easy to follow and understand (congratulation to authors).

I think data presented here about molecular testing is also valuable.

Some aspect needs to be improved, in order to do this manuscript much better.

Reviewer #2: Manuscript is well written with the required details regarding the study design and results.

This study provided up to date informations on antigen detection and PCR.

Results on antigen detection are of high importance since WHO recently recommend using this method in diagnosing PDH in people living with HIV.

When considering the required relevance of the discussion and simplification of the manuscript in general, it might have been better to submit experiments on antigen detection and PCR in two separate manuscripts.

PLOS authors have the option to publish the peer review history of their article (what does this mean?). If published, this will include your full peer review and any attached files.

Reviewer #1: Yes: Diego H. Caceres

Reviewer #2: No
---

## [Editor Report · Decision Letter 1]

6 Feb 2021

Dear Dr. Sifuentes-Osornio,

We are pleased to inform you that your manuscript 'Diagnostic accuracy of antigen detection in urine and molecular assays testing in different clinical samples for the diagnosis of progressive disseminated histoplasmosis in patients living with HIV/AIDS: A prospective multicenter study in Mexico.' has been provisionally accepted for publication in PLOS Neglected Tropical Diseases.

Best regards,

Angel Gonzalez, Ph.D.

Associate Editor

Todd Reynolds

Deputy Editor

---

## [Editor Report · Acceptance letter]

23 Feb 2021

Dear Prof. Sifuentes-Osornio,

We are delighted to inform you that your manuscript, "Diagnostic accuracy of antigen detection in urine and molecular assays testing in different clinical samples for the diagnosis of progressive disseminated histoplasmosis in patients living with HIV/AIDS: A prospective multicenter study in Mexico.," has been formally accepted for publication in PLOS Neglected Tropical Diseases.

Best regards,

Shaden Kamhawi

co-Editor-in-Chief

Paul Brindley

co-Editor-in-Chief
